# A PHASE TRANSITION INDUCES CATASTROPHIC OVERFITTING IN ADVERSARIAL TRAINING

## ABSTRACT

We derive the implicit bias of Projected Gradient Descent (PGD) Adversarial Training (AT). We show that a phase transition in the loss structure of as a function of the adversarial budget $\epsilon$ manifests as Catastrophic Overfitting (CO). Below a critical threshold $\epsilon_c$, single step methods efficiently provide an increase in robustness, while above this critical point, additional PGD steps and/or regularization are needed. We show that high curvature solutions arise in the implicit bias of PGD AT. We provide analytical and empirical evidence for our arguments by appealing to a simple model with one-dimensional inputs and a single trainable parameter, where the CO phenomenon can be replicated. In this model, we show that such high curvature solutions exist for arbitrarily small $\epsilon$. Additionally, we can compute the critical value $\epsilon_c$ in single-step AT for bounded parameter norms. We believe our work provides a deeper understanding of CO that aligns with the intuition the community has built around it.

## 1 INTRODUCTION

Machine Learning models are not by default robust to small corruptions in their input (Szegedy et al., 2014; Goodfellow et al., 2015). Adversarial Training (AT) (Madry et al., 2018) and its variants have proven to be one of the most effective strategies towards achieving adversarially robust models (Croce et al., 2020). Formulated as a min-max problem, Danskin's theorem can be invoked to solve AT via an alternating maximization-minimization procedure (Danskin, 1966; Latorre et al., 2023). Solving the inner maximization problem (usually with Projected Gradient Descent (PGD)) considerably slows down training, leading practitioners to use single-step approaches (Shafahi et al., 2019; Wong et al., 2020).

Nevertheless, when solving the inner maximization problem with a single (FGSM) step, AT presents a critical failure mode known as Catastrophic Overfitting (CO) (Wong et al., 2020; Andriushchenko and Flammarion, 2020), where for larger adversarial budgets $\epsilon$ the model overfits to be $100\%$ robust to (weak) single-step adversaries while being $0\%$ robust to (strong) multi-step adversaries.

It has been observed that multi-step AT leads to locally linear loss landscapes in the neighborhood of the training points (Andriushchenko and Flammarion, 2020). This has motivated the use of regularization methods to enforce local linearity and avoid CO in single-step AT (Qin et al., 2019; Moosavi-Dezfooli et al., 2019; Abad Rocamora et al., 2024). Despite the success of these methods and the efforts in understanding CO (Li et al., 2020a; Andriushchenko and Flammarion, 2020; Kim et al., 2021; Ortiz-Jimenez et al., 2023; He et al., 2023), little is known about why multi-step AT converges to locally linear solutions or which is the underlying phenomenon resulting in CO.

This work aims to fill this gap by connecting the empirical observations with a theoretical framework which allows us to quantitatively understand the onset of CO in AT.

To do so, we characterize the implicit bias of multi-step AT and link its terms with the well known high curvature and the appearance of CO (Andriushchenko and Flammarion, 2020). We construct a simple example where CO can be fully characterized and linked to a phase transition with a critical value $\epsilon_c$ above which CO appears. In this example, we demonstrate the voracity of our results in a controlled setting, where we explicitly find $\epsilon_c$. Furthermore, in this setting, we can show the failure modes of PGD AT, where CO can appear for arbitrarily small $\epsilon$ by increasing the curvature of the

---

**Algorithm 1** PGD Adversarial Training (AT) (Madry et al., 2018).

1: **Inputs:** Model weights $\boldsymbol{\theta}$, Dataset $\{\boldsymbol{x}_i, y_i\}_{i=1}^n$, # epochs $T$, # batches $M$, radius $\epsilon$, PGD step-sizes $\alpha_s$, learning rate $\gamma$ and initialization radius $\sigma$.
2: **for** $t \in [T]$ **do**
3:      **for** $i \in [M]$ **do**
4:          $\boldsymbol{\delta}_0^i = \text{Unif.}\left([-\sigma, \sigma]^d\right)$                                     ▷ Initialize perturbation
5:          **for** $s \in [S]$ **do**                                             ▷ Solving inner max
6:              $\boldsymbol{\delta}_s^i = \boldsymbol{\delta}_{s-1}^i + \alpha_s \cdot \epsilon \cdot \text{sign}\left(\nabla_{\boldsymbol{x}_i}\mathcal{L}(\boldsymbol{f_\theta}(\boldsymbol{x}_i + \boldsymbol{\delta}_{s-1}^i), y_i)\right)$ ▷ Signed gradient ascent step
7:              $\boldsymbol{\delta}_s^i = \min\{\max\{\boldsymbol{\delta}_s^i, -\epsilon\}, \epsilon\}$                      ▷ Project so that $\left|\left|\boldsymbol{\delta}_s^i\right|\right|_\infty \leq \epsilon$
8:      $\boldsymbol{\theta} = \boldsymbol{\theta} - \gamma \cdot \nabla_{\boldsymbol{\theta}}\mathcal{L}(\boldsymbol{f_\theta}(\boldsymbol{x}_i + \boldsymbol{\delta}_S^i), y_i)$                          ▷ SGD update

---

classifier, agreeing with intuition of the community in larger scale experiments. Lastly, we analyze the data re-scaling setting, where CO can be induced for smaller $\epsilon$ values.

We believe that our theoretical insights, jointly with our experimental results both in small and large scale models and datasets, provides a deeper understanding of CO and which factors contribute to its appearance/avoidance.

**Notation:** We use uppercase bold letters for matrices $\boldsymbol{X} \in \mathbb{R}^{m \times n}$, lowercase bold letters for vectors $\boldsymbol{x} \in \mathbb{R}^m$ and lowercase letters for numbers $x \in \mathbb{R}$. Accordingly, the $i^{\text{th}}$ row and the element in the $i, j$ position of a matrix $\boldsymbol{X}$ are given by $\boldsymbol{x}_i$ and $x_{ij}$ respectively. We use the shorthand $[n] = \{1, \cdots, n\}$ for any natural number $n$. We denote the indicator function as $\mathbb{1}[\cdot]$.

## 2 BACKGROUND

We analyze AT and its single-step variants in Section 2.1. In Section 2.2 we cover the efforts towards understanding and avoiding CO in single-step AT.

### 2.1 ADVERSARIAL TRAINING

AT can be formulated as a min-max optimization problem. Let $\{(\boldsymbol{x}_i, y_i)\}_{i=1}^n$ be the training dataset, with $\boldsymbol{x}_i \in \mathbb{R}^d$ and $y_i \in [o]$. Let $\boldsymbol{f_\theta} : \mathbb{R}^d \to \mathbb{R}^o$ be a classifier parameterized by $\boldsymbol{\theta} \in \mathbb{R}^p$, assigning a score to each class so that the predicted class is given by $\hat{y}_i = \arg\max_{j \in [o]} \boldsymbol{f_\theta}(\boldsymbol{x}_i)_j$. Let $\mathcal{L} : \mathbb{R}^o \times [o] \to \mathbb{R}^+$ be a loss function, the adversarial training problem can be formulated as:

$$\min_{\boldsymbol{\theta} \in \mathbb{R}^p} \frac{1}{n} \sum_{i=1}^n \max_{||\boldsymbol{\delta}_i||_\infty \leq \epsilon} \mathcal{L}(\boldsymbol{f_\theta}(\boldsymbol{x}_i + \boldsymbol{\delta}_i), y_i). \tag{AT}$$

For ease of notation, we define $\boldsymbol{\Delta} \equiv [\boldsymbol{\delta}_1, \cdots, \boldsymbol{\delta}_n]^\top$ and $g(\boldsymbol{\theta}, \boldsymbol{\Delta}) \equiv \frac{1}{n} \sum_{i=1}^n \mathcal{L}(\boldsymbol{f_\theta}(\boldsymbol{x}_i + \boldsymbol{\delta}_i), y_i)$. Then, Eq. (AT) becomes:

$$\min_{\boldsymbol{\theta} \in \mathbb{R}^p} \max_{\boldsymbol{\Delta}:||\boldsymbol{\delta}_i||_\infty \leq \epsilon} g(\boldsymbol{\theta}, \boldsymbol{\Delta}). \tag{1}$$

Eq. (AT) is usually solved by invoking Danskin's theorem Danskin (1966) to compute:

$$\nabla_{\boldsymbol{\theta}} \max_{\boldsymbol{\Delta}:||\boldsymbol{\delta}_i||_\infty \leq \epsilon} g(\boldsymbol{\theta}, \boldsymbol{\Delta}) = \nabla_{\boldsymbol{\theta}} g(\boldsymbol{\theta}, \boldsymbol{\Delta}^*), \quad \boldsymbol{\Delta}^* \in \arg\max_{\boldsymbol{\Delta}:||\boldsymbol{\delta}_i||_\infty \leq \epsilon} g(\boldsymbol{\theta}, \boldsymbol{\Delta}). \tag{2}$$

This allows one to solve Eq. (AT) as a minimization problem using first order methods, where at each iteration, the gradient is computed by (approximately) solving the inner maximization problem on the right-hand-side of Eq. (2). In order to solve the inner $\max$, Projected Gradient Descent (PGD) is commonly used (Madry et al., 2018). The standard AT procedure is covered in Algorithm 1. Note that in Line 6 of Algorithm 1, taking the gradient with respect to the perturbation $\boldsymbol{\delta}_{s-1}^i$ or the input $\boldsymbol{x}_i$ is equivalent, i.e., $\nabla_{\boldsymbol{x}_i}\mathcal{L}(\boldsymbol{f_\theta}(\boldsymbol{x}_i + \boldsymbol{\delta}_{s-1}^i), y_i) = \nabla_{\boldsymbol{\delta}_{s-1}^i}\mathcal{L}(\boldsymbol{f_\theta}(\boldsymbol{x}_i + \boldsymbol{\delta}_{s-1}^i), y_i)$.

The main drawback of AT is that solving the inner maximization problem considerably slows down training in comparison to standard training. This has motivated the use of single-step adversarial

training variants (Shafahi et al., 2019; Wong et al., 2020; Andriushchenko and Flammarion, 2020; de Jorge et al., 2022).

## 2.2 SINGLE-STEP AT AND CATASTROPHIC OVERFITTING

In order to alleviate the computational overhead introduced by solving the inner maximization problem, Shafahi et al. (2019) suggest alternating some maximization and minimization steps in a single batch and reducing the number of epochs. This is equivalent to moving line 8 in Algorithm 1 inside the PGD loop. Alternatively, a single-step (FGSM) attack (Goodfellow et al., 2015) can be employed. This is equivalent to setting $S = 1$ and $\alpha = 1$ in Algorithm 1.

Unfortunately, FGSM training often results in the so called *Catastrophic Overfitting* (CO) phenomenon (Wong et al., 2020). We define CO as follows:

**Definition 2.1** (($\beta, \eta$)-Catastrophic Overfitting (CO)). Let $\beta, \eta \in [0, 1]$. Let $\boldsymbol{f_\theta} : \mathbb{R}^d \to \mathbb{R}^o$ be a classifier trained with Algorithm 1 on a dataset $\{(\boldsymbol{x}_i, y_i)\}_{i=1}^n$ and adversarial budget $\epsilon$. The ($\beta, \eta$)-CO phenomenon is characterized by a high PGD accuracy and low robust accuracy:

$$\frac{1}{n} \sum_{i=1}^n \mathbb{1} \left[ \arg\max_{j \in [o]} f_{\boldsymbol{\theta}}(\boldsymbol{x_i} + \boldsymbol{\delta}_S^i)_j = y_i \right] \geq 1 - \beta, \quad \frac{1}{n} \sum_{i=1}^n \mathbb{1} \left[ \arg\max_{j \in [o]} f_{\boldsymbol{\theta}}(\boldsymbol{x_i} + \boldsymbol{\delta}_\star^i)_j = y_i \right] \leq \eta,$$

for some $\boldsymbol{\delta}_\star^i : \left\| \boldsymbol{\delta}_\star^i \right\|_\infty \leq \epsilon \quad \forall i \in [n]$ and $\boldsymbol{\delta}_S^i$ obtained with the same PGD hyperparameters as used for training with Algorithm 1. Intuitively, CO results in a model being robust to the PGD attacks seen during training, but not being robust to other $\ell_\infty$ perturbations bounded by $\epsilon$.

**Definition 2.2** (Critical adversarial budget $\epsilon_c$). Let $\beta, \eta \in [0, 1]$. Let $\boldsymbol{f_\theta} : \mathbb{R}^d \to \mathbb{R}^o$ be a classifier trained with Algorithm 1 on a dataset $\{(\boldsymbol{x}_i, y_i)\}_{i=1}^n$. If ($\beta, \eta$)-CO is observed for a given set of training hyperparameters and for all adversarial budgets $\epsilon > \epsilon_c$, but ($\beta, \eta$)-CO is not observed for any $\epsilon \leq \epsilon_c$, we denote $\epsilon_c$ as the critical adversarial budget.

It was initially thought that CO could be solved by adding uniform noise to the input image to add diversity in the gradient estimation (Wong et al., 2020; de Jorge et al., 2022). However, it was later shown that this only postponed the appearance of CO to larger $\epsilon$ (Andriushchenko and Flammarion, 2020; Abad Rocamora et al., 2024).

According to Andriushchenko and Flammarion (2020), multi-step AT converges to locally linear loss landscapes and this property is lost in single-step AT when CO appears. Conversely, regularizing the loss to be locally linear in a neighborhood of training points $\boldsymbol{x}_i$ for single-step AT can avoid CO. Let $\boldsymbol{x}_a$ and $\boldsymbol{x}_b$ be sampled uniformly from the ball $\{\boldsymbol{x}' : \|\boldsymbol{x} - \boldsymbol{x}'\|_\infty \leq \epsilon\}$, let $\boldsymbol{g}(\boldsymbol{x}) = \nabla_{\boldsymbol{x}} \mathcal{L}(\boldsymbol{f_\theta}(\boldsymbol{x}), y_i)$, local linearity can be measured and regularized through:

- GradAlign (Andriushchenko and Flammarion, 2020): $1 - \frac{\boldsymbol{g}(\boldsymbol{x})^\top \boldsymbol{g}(\boldsymbol{x}_a)}{\|\boldsymbol{g}(\boldsymbol{x})\|_2 \cdot \|\boldsymbol{g}(\boldsymbol{x}_a)\|_2}$

- CURE (Moosavi-Dezfooli et al., 2019): $\|\boldsymbol{g}(\boldsymbol{x}) - \boldsymbol{g}(\boldsymbol{x}_a)\|_2$

- LLR (Qin et al., 2019): $\left| \mathcal{L}(\boldsymbol{f_\theta}(\boldsymbol{x}_a), y) - \mathcal{L}(\boldsymbol{f_\theta}(\boldsymbol{x}), y) + (\boldsymbol{x}_a - \boldsymbol{x})^\top \boldsymbol{g}(\boldsymbol{x}) \right|$

- ELLE (Abad Rocamora et al., 2024), for $\alpha \in [0, 1]$:
$$\left| \mathcal{L}(\boldsymbol{f_\theta}(\alpha \cdot \boldsymbol{x}_a + (1 - \alpha) \cdot \boldsymbol{x}_b), y) - \alpha \cdot \mathcal{L}(\boldsymbol{f_\theta}(\boldsymbol{x}_a), y) - (1 - \alpha) \cdot \mathcal{L}(\boldsymbol{f_\theta}(\boldsymbol{x}_b), y) \right|.$$

Although these methods avoid CO in practice, there is no clear explanation on why multi-step AT converges to locally linear loss landscapes. In this work, we lay the groundwork towards answering this question.

Several works have tried to explain the appearance of CO. Kim et al. (2021) argue the loss landscape becomes "distorted" when CO appears, meaning that the original point $\boldsymbol{x}$ and the FGSM adversarial example $\boldsymbol{x}_{\text{FGSM}}$ are well classified. But, there exist misclassified points closer to $\boldsymbol{x}$, even in the convex combination of $\boldsymbol{x}$ and $\boldsymbol{x}_{\text{FGSM}}$. He et al. (2023) analyze CO from the "self fitting" perspective and argue that during FGSM training, the network encodes features into the gradient, relying less on the features of the image for classification and resulting in CO. He et al. (2023) also show that CO can happen in multi-step AT under large step sizes ($\alpha$ in Algorithm 1). Ortiz-Jimenez et al. (2023) conclude that CO can be induced by discriminative features that are relevant for classification, but not robustness.

Even though we can heuristically understand the potential factors leading to CO, the current theory is incomplete and not predictive, i.e., given a dataset, a network and AT hyperparameters, we cannot predict if CO will appear, or even quantitatively analyze it in an instructive way. In this work, we are able to understand this behavior in the language of implicit regularization, as well as fully characterize CO in a simple 1-dimensional model, and subsequently transfer these insights to more general settings.

## 3 THE IMPLICIT BIAS OF PGD AT

In this section we analyze the implicit bias of AT and its implications for CO and the loss landscape of the AT solutions. We include all of our proofs in Appendix E.

### 3.1 WHAT CAN PGD AT CONVERGE TO?

This question has been studied in-depth from the maximum margin perspective in Li et al. (2020b); Lv and Zhu (2022), which show that adversarially trained homogeneous networks converge in direction to a mixed-maximum-margin solution. Here, we take a different approach, focusing on the effective objective being optimized, and how does the $\epsilon$ perturbation affect the loss landscape.[1]. We are interested in understanding which quantities, apart from the loss at the unperturbed points, is AT implicitly minimizing. By Taylor-expanding the loss along the PGD adversarial perturbations from Algorithm 1, we arrive at the result in Proposition 3.1.

**Proposition 3.1** (The implicit bias of PGD AT). *Let $f_{\boldsymbol{\theta}} : \mathbb{R}^d \to \mathbb{R}^o$ be a twice-differentiable classifier trained with Algorithm 1, $\alpha_s = 1/S$, $\sigma = 0$ and the loss function $\mathcal{L} : \mathbb{R}^o \times [o] \to \mathbb{R}^+$ over the dataset $\{(\boldsymbol{x}_i, y_i)\}_{i=1}^n$. Let the input gradients and Hessians with respect to the input be $\boldsymbol{g}_{\boldsymbol{\theta}}(\boldsymbol{x}) = \nabla_{\boldsymbol{x}}\mathcal{L}(f_{\boldsymbol{\theta}}(\boldsymbol{x}), y) : \mathbb{R}^d \to \mathbb{R}^d$ and $\boldsymbol{H}_{\boldsymbol{\theta}}(\boldsymbol{x}) = \nabla^2_{\boldsymbol{x}\boldsymbol{x}}\mathcal{L}(f_{\boldsymbol{\theta}}(\boldsymbol{x}), y) : \mathbb{R}^d \to \mathbb{R}^{d \times d}$ respectively. We have that:*

$$\sum_{i=1}^n \mathcal{L}(f_{\boldsymbol{\theta}}(\boldsymbol{x}_i + \boldsymbol{\delta}_S^i), y_i) \approx \sum_{i=1}^n \left[ \mathcal{L}(f_{\boldsymbol{\theta}}(\boldsymbol{x}_i), y_i) + \sum_{j=1}^S \left( \frac{\epsilon}{S} \left\| \boldsymbol{g}_{\boldsymbol{\theta}}(\boldsymbol{x}_i + \boldsymbol{\delta}_{j-1}^i) \right\|_1 \right. \right. \tag{3}$$

$$\left. \left. + \frac{\epsilon^2}{2S^2} \operatorname{sign}(\boldsymbol{g}_{\boldsymbol{\theta}}(\boldsymbol{x}_i + \boldsymbol{\delta}_{j-1}^i))^\top \boldsymbol{H}_{\boldsymbol{\theta}}(\boldsymbol{x}_i + \boldsymbol{\delta}_{j-1}^i) \operatorname{sign}(\boldsymbol{g}_{\boldsymbol{\theta}}(\boldsymbol{x}_i + \boldsymbol{\delta}_{j-1}^i)) \right) \right]$$

$$= \mathcal{L}_0(\boldsymbol{\theta}) + \frac{\epsilon}{S} \sum_{i=1}^n \sum_{j=1}^S \mathcal{R}_{1;i,j} + \frac{\epsilon^2}{2S^2} \sum_{i=1}^n \sum_{j=1}^S \mathcal{R}_{2;i,j}.$$

Here, $\mathcal{L}_0(\boldsymbol{\theta})$ is the unperturbed loss, and $\mathcal{R}_{1;i,j} \equiv \left\| \boldsymbol{g}_{\boldsymbol{\theta}}(\boldsymbol{x}_i + \boldsymbol{\delta}_{j-1}^i) \right\|_1, \mathcal{R}_{2;i,j} \equiv \operatorname{sign}(\boldsymbol{g}_{\boldsymbol{\theta}}(\boldsymbol{x}_i + \boldsymbol{\delta}_{j-1}^i))^\top \boldsymbol{H}_{\boldsymbol{\theta}}(\boldsymbol{x}_i + \boldsymbol{\delta}_{j-1}^i) \operatorname{sign}(\boldsymbol{g}_{\boldsymbol{\theta}}(\boldsymbol{x}_i + \boldsymbol{\delta}_{j-1}^i))$ are the implicit regularizer and bias terms, for $i \in [n]$ and $j \in [S]$. By minimizing the $\mathcal{R}_{1;i,j}$ term, we take the norm of the gradients along the PGD trajectory to be zero, which in general helps with robustness because, intuitively, if $\left\| \boldsymbol{g}_{\boldsymbol{\theta}}(\boldsymbol{x}_i + \boldsymbol{\delta}) \right\|_1 = 0, \forall \boldsymbol{\delta} : \left\| \boldsymbol{\delta} \right\|_\infty \leq \epsilon$ and the loss at $\boldsymbol{x}_i$ is low, then the classifier is robust. The $\mathcal{R}_{2;i,j}$ term corresponds to the second order directional derivatives, which AT implicitly regularizes by taking curvature to zero (Andriushchenko and Flammarion, 2020) and popular single-step AT methods regularize its norm uniformly through approximations (Qin et al., 2019; Abad Rocamora et al., 2024).

Nevertheless, in Proposition 3.1, we observe the second order directional derivatives, not their absolute value. Then, theoretically, one solution to minimizing the implicit bias, would be to take second order directional derivatives to $-\infty$. This does not happen in practice for multi-step AT, as multi-step AT converges to have small curvature in the neighborhood of training points. However, some questions arise: *(i) Can we have PGD AT solutions where curvature is high? (ii) Can we have solutions where CO appears for any $S$ and $\epsilon$? (iii) Can we understand why such solutions do not appear in practice?*

---

[1]While we did not study the implicit bias from the perspective of directional convergence to a margin, we believe the study of CO can also be phrased in this way, showing a mismatch between mixed-margin solutions with $\epsilon > \epsilon_c$.

To answer question *(i)*, we note that the interplay between $\mathcal{L}_0(\boldsymbol{\theta}), \mathcal{R}_{1;i,j}$ and $\mathcal{R}_{2;i,j}$ strongly depends on the details of the data and architecture, but can be characterized by the scalar coefficients $\epsilon/S, \epsilon^2/2S^2$. In particular, solutions can exist which it is preferable to minimize $\mathcal{R}_{2;i,j}$ over the other terms. This case would lead to CO in PGD AT, where $\mathcal{R}_{2;i,j}$ obtains large negative values.

In order to tackle question *(ii)*, in Proposition 3.2 we present a mechanism that can induce arbitrarily small $\epsilon_c$ for which CO appears in more complex models, where the only assumption is that the first layer is affine.

**Proposition 3.2** (Re-scaling the data re-scales $\epsilon$)**.** *Given a classification dataset $\{(\boldsymbol{x}_i, y_i)\}_{i=1}^n$ and a model $\boldsymbol{f_\theta} : \mathbb{R}^d \to \mathbb{R}^o$ where the first layer is affine, i.e., $\boldsymbol{f_\theta}(\boldsymbol{x}) = \hat{\boldsymbol{f}}_{\hat{\boldsymbol{\theta}}}(\boldsymbol{W}\boldsymbol{x} + \boldsymbol{b})$, where $\hat{\boldsymbol{f}}_{\hat{\boldsymbol{\theta}}}$ are all the layers except the first one and $\boldsymbol{\theta} = \hat{\boldsymbol{\theta}} \cup \{\boldsymbol{W}, \boldsymbol{b}\}$. Let $\alpha \in \mathbb{R} \setminus \{0\}$. Solving Eq. (AT) in a re-scaled dataset $\{(\alpha \cdot \boldsymbol{x}_i, y_i)\}_{i=1}^n$ with adversarial budget $\epsilon_\alpha$ is equivalent to solving Eq. (AT) in the standard dataset $\{(\boldsymbol{x}_i, y_i)\}_{i=1}^n$ with adversarial budget $\epsilon = \epsilon_\alpha/\alpha$.*

**Corollary 3.3** (Re-scaled datasets present re-scaled $\epsilon_c$)**.** *If training with Algorithm 1 in the dataset $\{(\boldsymbol{x}_i, y_i)\}_{i=1}^n$ presents a critical adversarial budget $\epsilon_c$ as in Definition 2.2, training in the re-scaled dataset $\{(\boldsymbol{x}_i \cdot \alpha, y_i)\}_{i=1}^n$ will present a critical adversarial budget $\epsilon_{c,\alpha} = \alpha \cdot \epsilon_c$.*

With Proposition 3.2 and Corollary 3.3, we have a mechanism to re-scale the dataset and produce smaller $\epsilon_c$ that applies to modern deep architectures like ResNets (He et al., 2016) and any training dataset. In particular, Corollary 3.3 shows that the scale of $\epsilon_c$ does not only depend on the hyperparameters of Algorithm 1, but also the scale of the dataset. In Section 5.5 we confirm the result holds for ResNets trained in MNIST, SVHN and CIFAR10.

Lastly, regarding question *(iii)*, CO can be found for large $S$ under certain step-sizes $\alpha_s$ (He et al., 2023) and our experimental results in Section 5.4 and Appendix F.1 with $S = 2$ show curvature indeed goes to large values. Nevertheless, if $S$ is large, $\alpha_s$ is chosen appropriately and some engineering tricks like adding noise prior to the PGD attack (Wong et al., 2020; de Jorge et al., 2022) or weight decay (Goodfellow et al., 2016) are performed, PGD AT is understood to provide reliable solutions (Croce and Hein, 2020b;a). We believe some key aspects are noise and weight decay. With noise before the PGD attack, we can affect the PGD trajectory and avoid solutions like the ones presented in Theorem 4.1. Similarly, by adding weight decay, the norm of the parameters is constrained, which in our toy model directly controls curvature and CO (Corollary 4.3). for additional discussion on these topics, we refer to Appendix C and Section 4.2

## 4 CATASTROPHIC OVERFITTING AS A PHASE TRANSITION

In this section, we construct a solvable model in which the effective loss in Proposition 3.1 can be written in full, as well as in its approximate form. Studying this model allows us to explain why CO occurs abruptly at a critical perturbation threshold value $\epsilon_c$ which can be exactly computed, building further intuition into what transpires in real-world settings.

### 4.1 TOY MODEL

Here, we characterize the transition from robust generalization to CO as a phase transition (PT) in the perturbation value, and highlight the connections between curvature and over-fitting.

Consider a training dataset consisting of two points $x_1 = -x_2 \in \mathbb{R}$. To illustrate the effect of AT, we may choose these points to be $x_{1,2} = \pm\pi/2$. The task is binary classification, with labels generated by taking $y = \Theta(\sin(x))$, where $\Theta(z)$ is the Heaviside function. We use the cross-entropy (CE) loss $\langle\mathcal{L}\rangle = \frac{1}{2}\sum_{i=1}^2 \mathcal{L}(f_\theta(x_i), y_i)$, where $\mathcal{L}(f_\theta(x_i), y_i) = -y_i \log p_i - (1-y_i)\log(1-p_i)$, and the probabilities are simply $p_i = e^{f_\theta(x_i)}/(1 + e^{f_\theta(x_i)})$. We take the network function to be

$$f_\theta(x_i) = \sin(\theta x_i), \qquad \theta \in [0, \theta_{\max}], \tag{4}$$

where $\theta$ is the single network parameter. Note that the network weight is bounded by $\theta_{\max}$, which we take to be $\theta_{\max} = 10$ for the rest of this section. This is done to match real-world settings in which weights do not diverge, while at the same time consisting a large enough value interval to observe CO. We provide a detailed analysis of this toy model trained using AT in Appendix A, and employ the results in the rest of the section.

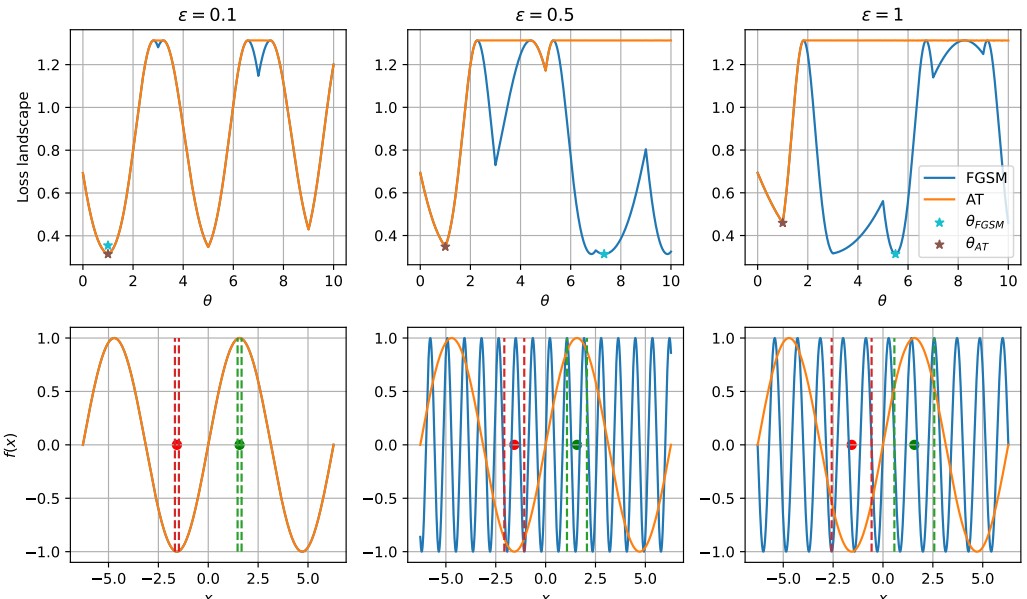

Figure 1: **Loss landscape and optimal classifiers for the toy model and** $\epsilon \in \{0.1, 0.5, 1\}$**:** In the *top row*, we plot the loss function with respect to the single parameter in the model $\theta$. In the *bottom row*, we plot the optimal classifiers according to the loss in the upper row, the two training points (🔴 and 🟢) and their adversarial regions (-- and --). The optimal classifier obtained with FGSM (★) coincides with one obtained with AT (★) for $\epsilon = 0.1$. For larger values of $\epsilon$, the FGSM and AT solutions differ, with FGSM obtaining distorted solutions where the FGSM-attacked points are well classified, but clean samples are not.

The effective loss, which is optimized using the one step AT, is simply $\mathcal{L}(f_\theta(x_i + \delta_i), y_i)$, which is given explicitly by

$$\langle \mathcal{L} \rangle = \frac{1}{2} \sum_{i=1}^{2} \mathcal{L}(f_\theta(x_i + \delta_i), y_i) = \log \left( 1 + e^{-\sin\left(\theta\left(\frac{\pi}{2} - \epsilon \, \text{sign}\left(\theta \cos\left(\frac{\theta\pi}{2}\right)\right)\right)\right)} \right). \tag{5}$$

The loss in Eq. (5) exhibits interesting properties when varying the perturbation $\epsilon$. When $\epsilon = 0$, the loss has infinitely many degenerate minima, at $\theta_{\min} = 4n + 1$, and as many degenerate maxima at $\theta_{\max} = 4n + 3$ where $n \in \mathbb{Z}^+$. While these extrema share the same loss values, they differ in loss curvature with respect to the samples $x_i$, measured by the sample-wise Hessian

$$H(\theta) \equiv \partial_{x_i}^2 \mathcal{L}(f_\theta(x_i), y_i)|_{x_i = \pm \frac{\pi}{2}} = \frac{\theta^2 \left( \sin\left(\frac{\pi\theta}{2}\right) + e^{\sin\left(\frac{\pi\theta}{2}\right)} \left( \sin\left(\frac{\pi\theta}{2}\right) + \cos^2\left(\frac{\pi\theta}{2}\right) \right) \right)}{2 \left( 1 + e^{\sin\left(\frac{\pi\theta}{2}\right)} \right)^2} \propto \theta^2, \tag{6}$$

which increases with $\theta$, implying that the lowest curvature solution is at $\theta_{\min} = 1$, which is indeed the solution for $\epsilon \to 0$. However, as $\epsilon$ is increased, the loss landscape changes, and certain minima obtain higher or lower loss values, changing both the local and global minima. In particular, at $\epsilon_c = \pi/8$, the minimum at $\theta = 1$ becomes unstable, and the loss is instead minimized at $\theta_{\min} = 7$, which was originally a maximum of the loss. We explicitly show that this occurs for the toy model in Fig. 2 (**b**).

The abrupt transition of the system from one minimum to another is precisely the behavior observed in physical systems which undergo a first order phase transition, whose theory is well understood (Landau and Lifshits, 1958). These transitions exhibit a discontinuity in the first derivative of the free energy with respect to some thermodynamic variable, analogized here by the discontinuity of the derivative of the loss with respect to $\theta$ at the transition point $\epsilon_c$. We show how the loss landscape evolves with $\epsilon$ in Fig. 1, as well as the solutions found by the network $f(x)$. Clearly, for small $\epsilon < \epsilon_c$, the system is in a "generalizing" phase, where the curvature is low and the optimal robust

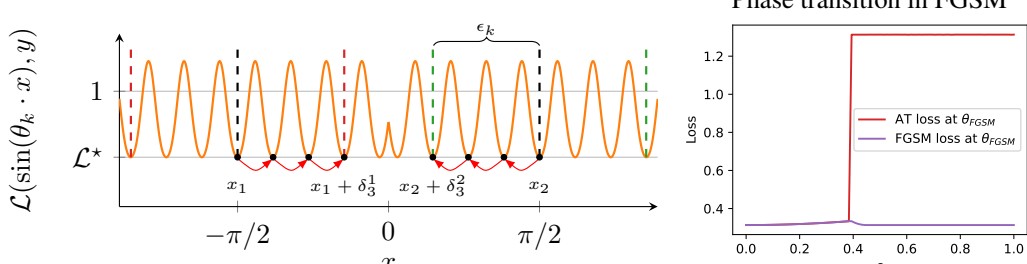

Figure 2: **(a) Visualization of Theorem 4.1 for** $S = 3, a = 1{,}000$ **and** $k = 4$**:** For any $S$, we can choose $a$ and $k$ so that $\theta_k$ is as accurate as desired and CO appears for $\epsilon_k$ arbitrarily small. **(b) Phase transition in the toy model:** The critical value is $\epsilon_c = \pi/8$. We evaluate the FGSM and AT losses at the optimal weight for the FGSM loss, i.e., $\theta_{\text{FGSM}}$. For $\epsilon > \epsilon_c$, the phase transition occurs, resulting in a sudden increase in the AT loss and a decrease in the FGSM loss.

solution is obtained. However, for $\epsilon \geq \epsilon_c$, the model is minimized at an "overfitting" phase, where the perturbed data points are overfit by favoring a high curvature solution, completely failing on the original data. This example illustrates that the essence of Catastrophic Overfitting lies in the phase transition from a low curvature to a high curvature region of the solution.

In order to connect this point to Proposition 3.1, we argue it is sufficient to study the first few terms in the series expansion of Eq. (5) with respect to $\epsilon$, in the $\epsilon \ll 1$ limit, given by

$$\langle \mathcal{L}(\theta, \epsilon) \rangle = \log \left( 1 + e^{-\sin\left(\frac{\pi\theta}{2}\right)} \right) + \epsilon \left| \frac{\theta \cos\left(\frac{\pi\theta}{2}\right)}{1 + e^{\sin\left(\frac{\pi\theta}{2}\right)}} \right| + \epsilon^2 \, \text{sign} \left( \theta \cos\left(\frac{\pi\theta}{2}\right) \right)^2 H(\theta) + \mathcal{O}(\epsilon^3), \quad (7)$$

where the first term corresponds to the unperturbed loss $\langle \mathcal{L}(\theta, 0) \rangle$, the second term is the sum of $L_1$ norms of the sample gradients, and the last term is a product of the sample wise Hessian and the sign of the sample gradients. In Fig. 1, we show how Eq. (7) approximates the full solution, and find that it predicts the same PT, but at a slightly lower $\epsilon$ value. We can estimate $\epsilon_c$ by requiring that the maximum at $\theta_{\max} = 7$ becomes a minimum, i.e., $\langle \mathcal{L}(1, \epsilon) \rangle = \langle \mathcal{L}(7, \epsilon) \rangle$, resulting in the value $\epsilon_c = \frac{1}{7} \sqrt{2 + \frac{2}{e}} \simeq 0.24$.

## 4.2 MULTI-STEP AT IN THE TOY MODEL

In Theorem 4.1 and Corollary 4.2, we present an example where CO appears for arbitrarily small $\epsilon$ and arbitrarily large $S$. We then analyze the interplay between the norm of parameters of the model $|\theta|$ and the number of steps $S$ in the appearance of CO. Finally, we discuss how CO can be induced in deep models by re-scaling the dataset.

**Theorem 4.1** (Solutions of the PGD AT problem). *Let the classifier defined in Section 4.1 be trained with Algorithm 1, $S$ steps and PGD step sizes $\alpha_s = 1/S$, $\forall s = 1, \cdots, S$. Let $a > 0$ and $b_k = \frac{1 + 4 \cdot k}{1 - \frac{1}{a}}$ for $k = 1, \cdots, \infty$, the pair*

$$\theta_k = b_k, \; \epsilon_k = \frac{2\pi S}{b_k}$$

*gives us weights $\theta_k$ for the corresponding adversarial budget $\epsilon_k$, so that:*

$$\frac{1}{2} \sum_{i=1}^{2} \mathcal{L}(f_{\theta_k}(x_i + \delta_S^i), y_i) - \mathcal{L}^\star \leq \pi \frac{b_k}{a},$$

*where $\mathcal{L}^\star = \log(1 + e) - 1$ is the optimal loss for a classifier defined as in Section 4.1.*

**Corollary 4.2** $((0,0)$-CO exists at arbitrarily small $\epsilon$ for any $S$). *Given any $S$, by increasing $a$ and $k$ we can take $\epsilon_k$ arbitrarily close to zero with arbitrarily accurate solutions $\theta_k$, where the points $x_i \pm \frac{\epsilon_k}{2S}$ are misclassified.*

**Corollary 4.3** (Bounding $|\theta|$ and increasing $S$ can avoid CO). *Let $b_k$ be constrained as $|b_k| \leq B$, we have that $\epsilon_k \geq \frac{2\pi S}{B}$. Meaning that bonding the norm of $\theta$ and increasing the number of steps can help avoid CO by avoiding the solutions in Theorem 4.1.*

*Remark* 4.4 (Additional CO solutions). In Theorem 4.1 we present sufficient conditions to observe CO. Nevertheless, other solutions not captured by Theorem 4.1 can be observed, see Fig. 1

In Theorem 4.1 we provide a constructive mechanism to build PGD AT solutions where CO appears for arbitrarily small $\epsilon$ Corollary 4.2. This is depicted in Fig. 2 **(a)**, where for $S = 3$, $a = 1,000$, setting $k = 4$ results in $\epsilon_k \approx 1.1077$ and the solution $\theta_k \approx 16.98301$, for which both 3-step PGD points $(x_i + \delta_3^i)$ are well classified, but the true adversarial accuracy is zero because of Corollary 4.2.

In Corollary 4.3, we show that by looking at the ratio defining $\epsilon_k = \frac{2\pi S}{\theta_k}$, it is clear that bounding the norm of the parameters can lowerbound $\epsilon_k$ and increasing $S$ will improve such lowerbound. This avoids the solutions constructed with Theorem 4.1 and overall helps avoiding CO. In Section 5.3, we observe a similar phenomenon in real world datasets and models.

## 5 EXPERIMENTS

In Section 5.1 we present our experimental setup. In Section 5.2 we numerically analyze the loss landscape of AT and FGSM on the toy model, and characterize their global minima. Next, in Section 5.3, we demonstrate the phase transition ocurs for small $S$ in real world image classification datasets and models. Finally, in Section 5.5, we demonstrate our findings from Proposition 3.2 and Corollary 3.3 in image classification experiments.

### 5.1 EXPERIMENTAL SETUP

For the image classification experiments, we use the popular MNIST (LeCun et al., 1998), SVHN (Netzer et al., 2011) and CIFAR10 (Krizhevsky, 2009) datasets. We train our models with Stochastic Gradient Descent, momentum 0.9, weight decay 0.0005, batch size 128 and the standard schedule with cyclic learning rate proposed by (Andriushchenko and Flammarion, 2020), with 15 epochs for MNIST/SVHN and 30 epochs for CIFAR10. We employ the PreActResNet18 architecture (He et al., 2016) and $\sigma = 0$ in Algorithm 1. in all of our image classification experiments. For different learning rate schedules like (Rice et al., 2020) and architectures like ViT-Small (Dosovitskiy et al., 2021), we refer to Appendix F. All of our training setups are repeated over 3 random seeds to report the average performance and confidence. All of our experiments are conducted on a single machine with an NVIDIA A100 SXM4 40 GB GPU.

### 5.2 CATASTROPHIC OVERFITTING IN THE TOY MODEL

For our toy model (Section 4.1), we can obtain a closed form expression for the effective loss function which is optimized by single-step AT with respect to $\theta$, see Eq. (5). Unfortunately, in order to compute the AT landscape, we would have to exactly compute for every $\theta$:

$$\delta_\star^i = \underset{\delta_i \in [-\epsilon, \epsilon]}{\arg\max} \ \mathcal{L}(\sin(\theta \cdot (x_i + \theta_i)), y_i). \tag{8}$$

As an alternative, we evaluate the loss at $10,000$ evenly distributed $\theta$ values in the $[0, 10]$ interval. For every $\theta$ and training sample $(x_i, y_i)$, we evaluate the loss at 100 evenly distributed $\theta_i$ values in the $[-\epsilon, \epsilon]$ interval, and take the maximum over those in order to estimate Eq. (8). In Fig. 1 we can observe that the FGSM solutions for $\epsilon \in \{0.5, 1\}$ display a distorted landscape, coinciding with the findings of Kim et al. (2021) for larger networks and datasets.

In order to compute the critical value $\epsilon_c$, we repeat the procedure for 100 evenly spaced values of $\epsilon$ in the $[0, 1]$ interval.

In Fig. 8-(b), we report the FGSM and AT losses evaluated at the FGSM solution ($\theta_{\text{FGSM}}$). We find $\epsilon_c \approx 0.3927$, above this value, the AT loss suddenly increases and the FGSM loss starts decreasing. The latter should not be observed as the true adversarial loss monotonically increases with $\epsilon$.

### 5.3 THE PHASE TRANSITION AT A LARGER SCALE

We train PreActResNet18 with the ReLU activation function with Algorithm 1, $S \in \{1, 2, 3, 4\}$ and $\alpha_s = \epsilon/S$. We scan over $\epsilon$ values from 0 to $70/255$, $12/255$ and $16/255$ for MNIST, SVHN and

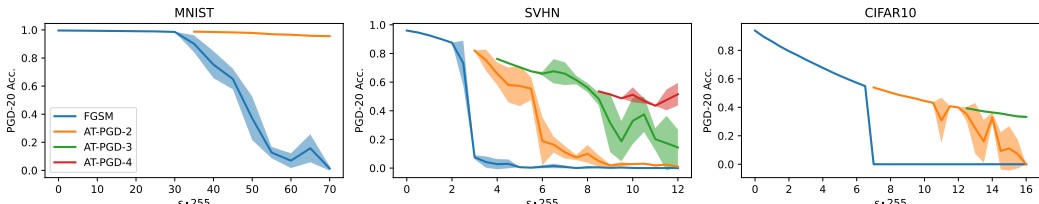

Figure 3: **The phase transition in image classification:** We train PreActResNet18 with $1, 2, 3$ and $4$ PGD steps from the first $\epsilon$ value where CO appears for one PGD step less. Larger $\epsilon$ values require more and more steps to not present CO.

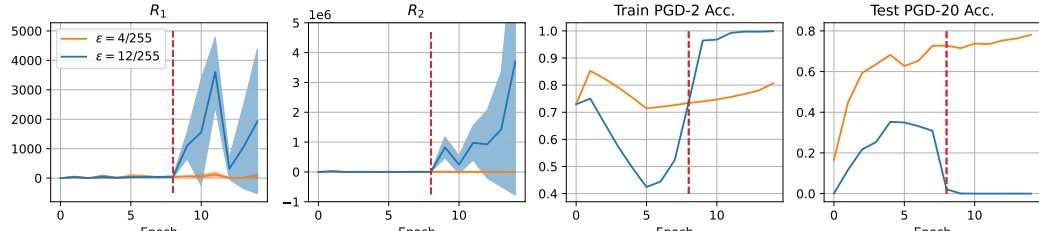

Figure 4: **Implicit bias terms in PreActResNet18-swish trained with $S = 2$ on SVHN:** We train with $\epsilon \in \{4, 12\}/255$ and report $\mathcal{R}_1$ and $\mathcal{R}_2$ from Proposition 3.1, the training PGD-2 accuracy and test PGD-20 accuracy. For $\epsilon = 12/255$, the implicit bias terms explode at the same time as CO occurs, which we mark with a vertical dashed red line (**- -**).

CIFAR10 respectively. For each $S$, we start training at the first $\epsilon$ value that presented CO for $S - 1$ steps. We report the average final PGD-20 test accuracy over 3 random seeds for every $\epsilon$ value.

In Fig. 3 we can observe that the phase transition leading to CO is observed in MNIST, SVHN and CIFAR10 for progressively larger $\epsilon_c$ for larger $S$. The phase transition is depicted by the abrupt decay of the PGD-20 Acc. to zero in a small $\epsilon$ change. This align with our results in the toy model despite the simplistic assumptions Theorem 4.1 and Corollary 4.3.

Our analysis in the toy model covers the existence of a CO solution with lower loss than the robust solution for larger $\epsilon$. Recent works argue longer training schedules might lead to CO (Kim et al., 2021; Abad Rocamora et al., 2024). According to our analysis this can be the case, as longer schedules might converge to the CO solutions shorter schedules did not. In Appendix F.2 we find that $\epsilon_c$ can be slightly smaller in certain datasets for longer schedules.

### 5.4 EMPIRICAL VALIDATION OF SECTION 3 FOR DEEP MODELS

In this section, we validate our hypothesis from Section 3, i.e., that curvature explodes in the points along the PGD trajectory, leading to CO. We train PreActResNet18 with the swish activation. This is necessary as the model needs to be twice-differentiable to compute $\mathcal{R}_1$ and $\mathcal{R}_\in$ from Proposition 3.1, which we compute on the first 3 datapoints of the training set after every epoch. We train on the SVHN dataset using Algorithm 1 with $S = 2$ and $\alpha_1 = \alpha_2 = 1/S$. In order to compare the behavior without and with CO, we select $\epsilon \in \{4, 12\}/255$ following our insights from Section 5.3.

In Fig. 4, we can observe that for the model trained with $\epsilon = 4/255$, we have that both $\mathcal{R}_1$ and $\mathcal{R}_2$ remain stable and close to $0$ and the training and test adversarial accuracies consistently grow. Alternatively, or $\epsilon = 12/255$, the implicit bias terms explode at the same time as CO occurs. Deviating from our hypothesis in Section 3, $\mathcal{R}_1$ does not remain close to $0$ and $\mathcal{R}_2$ goes to $+\infty$ instead of $-\infty$.

### 5.5 RE-SCALING FOR OBTAINING SMALLER $\epsilon_c$

In this section, we empirically validate the results from Proposition 3.2 and Corollary 3.3. To do so, we train in the modified dataset $\{(\boldsymbol{x}_i \cdot \alpha, y_i)_{i=1}^n\}$ with $\alpha \in \{0.25, 0.5, 0.75\}$, with $\alpha = 1$ being the

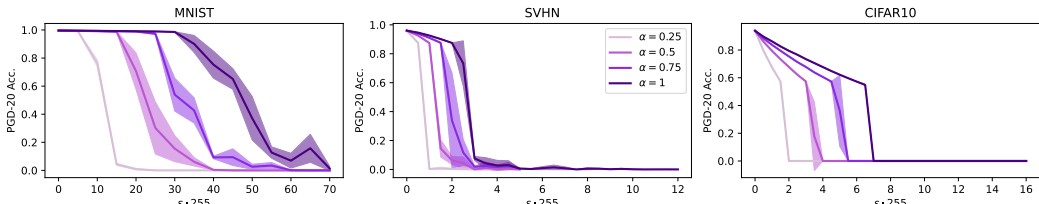

Figure 5: **Phase transition in re-scaled datasets with FGSM:** We preprocess the datasets by multiplying the inputs $x$ by $\alpha \in \{0.25, 0.5, 0.75, 1\}$. Re-scaling the dataset inputs by a factor of $\alpha$ produces proportionally smaller $\epsilon_c$ in single-step AT.

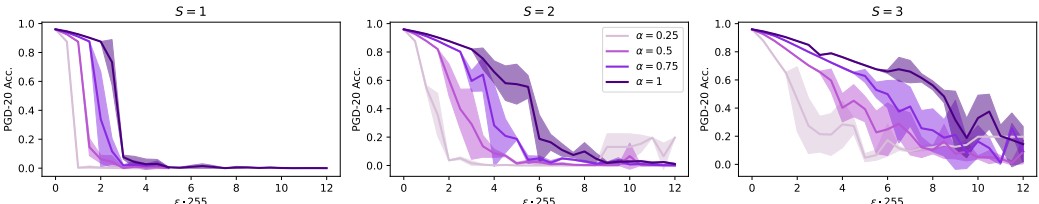

Figure 6: **Phase transition in re-scaled SVHN with multiple steps:** We preprocess the datasets by multiplying the inputs $x$ by $\alpha \in \{0.25, 0.5, 0.75, 1\}$ and train with $S \in \{1, 2, 4\}$ PGD steps. Even with larger $S$, CO can be produced earlier by re-scaling the training dataset.

original dataset. First, we train in MNIST, SVHN and CIFAR10 with $S = 1$. Then, we focus in the SVHN dataset, where CO can happen for $S \in \{1, 2, 3\}$ (See Fig. 3) in order to better analyze how the critical values of epsilon are displaced for larger $S$.

In Fig. 5, we can observe a clear proportionality in $\epsilon_c$ with respect to $\alpha$, with CO appearing earlier the smaller the $\alpha$. Similarly, for any number of steps $S$ for SVHN, the critical values $\epsilon_c$ are re-scaled accordingly, confirming the result in Corollary 3.3 and showing that CO can be induced for small $\epsilon$.

## 6 CONCLUSION

In this work, we contribute to a deeper understanding of catastrophic overfitting in adversarial training by linking its appearance with a phase transition, where a CO solution attains a lower loss than the robust solution for $\epsilon$ larger than a critical value $\epsilon_c$. To do so, we derive the implicit bias of PGD AT and show that high negative curvature solutions are characteristic of CO and can appear even in multi-step PGD. We propose a toy model where the phase transition can be fully analyzed and the high curvature solutions can be analytically obtained. We additionally provide a scaling argument for how the value of $\epsilon_c$ can change by re-scaling the dataset.

**Future work and limitations:** Our theoretical insights from the toy model show that by constraining the parameter norm, we can avoid high curvature solutions with CO, see Corollary 4.3. Nevertheless, this result does not necessarily extend to deep, more complex models.

Based on our implicit bias analysis, avoiding high curvature solutions is key to avoiding CO. Moreover, our experimental results in Appendix F.3 and the results of Singla et al. (2021) indicate that architectures which have intrinsically low curvature can avoid CO. This motivates the need of a theory for linking CO with curvature in deep models.

In this work, we conduct our theoretical and experimental analysis without noise prior to the PGD attack, i.e., $\sigma = 0$ in Algorithm 1. Using $\sigma > 0$ has been shown to help increase $\epsilon_c$ with $S = 1$ (Wong et al., 2020; de Jorge et al., 2022). In multi-step AT, $\sigma = \epsilon$ is used in practice (Madry et al., 2018). We believe that understanding the behavior of Algorithm 1 with $\sigma > 0$ is an interesting avenue and our insights with $\sigma = 0$ can foster such developments.

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

## CONTENTS OF THE APPENDIX

In Appendix A we provide additional derivations for our toy model. Then, in Appendices B and C we further analyze the implicit bias of AT in the simple case of $S = 1$ and $S = 2$. Next, in Appendix E we present our proofs and finally, in Appendix F we present additional experimental results not fitting in the manuscript.

## A  TOY MODEL FOR CATASTROPHIC OVERFITTING

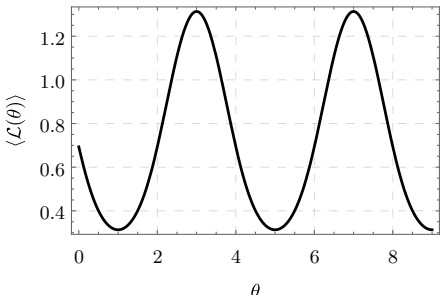 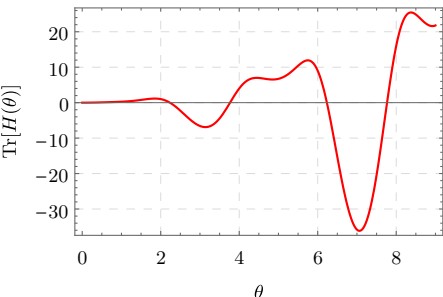

Figure 7: **Loss landscape and sample Hessian for the toy model and $\epsilon = 0$:** On the *left*, we plot the loss function with respect to the single parameter in the model $\theta$. On the *right*, we plot the Hessian with respect to the samples. As described in the main text, the loss has infinite degenerate extrema, where the curvature term, proportional to sample Hessian, grows as $\theta^2$.

Here, we provide a detailed analysis of the toy model presented in the main text, under a single step adversarial attack.

Consider a training dataset consisting of two points $x_1 = -x_2 \in \mathbb{R}$, where we choose $x_{1,2} = \pm\pi/2$. The task is binary classification, with labels generated by taking $y = \Theta(\sin(x))$, where $\Theta(z)$ is the Heaviside function. We use the cross-entropy (CE) loss $\langle\mathcal{L}\rangle = \frac{1}{2}\sum_{i=1}^{2}\mathcal{L}(f_\theta(x_i), y_i)$, where $\mathcal{L}(f_\theta(x_i), y_i) = -y_i \log p_i - (1 - y_i)\log(1 - p_i)$, and the probabilities are simply $p_i = e^{f_\theta(x_i)}/(1 + e^{f_\theta(x_i)})$. We take the network function to be

$$f_\theta(x_i) = \sin(\theta x_i), \qquad \theta \in \mathbb{R}^+. \tag{9}$$

where $\theta$ is the single network parameter.

The loss is given by the mean over the single sample losses taken on the two pairs of samples and labels $(x_1, y_1) = (x_1, 1)$ and $(x_2, y_2) = (x_2, 0)$, as

$$\langle\mathcal{L}\rangle = \frac{1}{2}\sum_{i=1}^{2}\mathcal{L}(f_\theta(x_i), y_i) = \frac{1}{2}\left(-\log\left(\frac{1}{1 + e^{-\sin(\theta x_1)}}\right) - \log\left(\frac{1}{1 + e^{\sin(\theta x_2)}}\right)\right) \tag{10}$$

$$= -\log\left(\frac{1}{1 + e^{-\sin\left(\frac{\pi\theta}{2}\right)}}\right).$$

This loss is minimized when $\partial_\theta\langle\mathcal{L}\rangle = 0$, which gives $\cos\left(\frac{\pi\theta}{2}\right) = 0$, satisfied for the set of degenerate minima $\theta_{\min} = 4n + 1$, and a set of maxima at $\theta_{\max} = 4n + 3$ where $n \in \mathbb{Z}^+$. Under an FGSM attack with parameter $\epsilon$, the effective loss being optimized is $\mathcal{L}(f_\theta(x_i + \delta_i), y_i)$, where

$$\delta_t^i = \epsilon \cdot \text{sign}\left(\nabla_{x_i}\mathcal{L}(f_{\theta_t}(x_i), y_i)\right) = \epsilon\,\text{sign}\left(\begin{array}{c} -\dfrac{\theta\cos(\theta x_1)}{2e^{\sin(\theta x_1)} + 2} \\[2mm] \dfrac{\theta\cos(\theta x_2)}{2e^{-\sin(\theta x_2)} + 2} \end{array}\right), \tag{11}$$

and is given explicitly by

$$\langle\mathcal{L}\rangle = \frac{1}{2}\sum_{i=1}^{2}\mathcal{L}(f_\theta(x_i + \delta_i), y_i) = \log\left(1 + e^{-\sin\left(\theta\left(\frac{\pi}{2} - \epsilon\,\text{sign}\left(\theta\cos\left(\frac{\theta\pi}{2}\right)\right)\right)\right)}\right). \tag{12}$$

The extrema landscape of the pertrubed loss differs from the original loss, increasingly so as $\epsilon$ is increased. While the perturbed loss can be fully described numerically, in order to gain analytical insights, it is worthwhile to Taylor expand the loss function for $\epsilon \ll 1$ as

$$\langle \mathcal{L}(\theta, \epsilon) \rangle = \frac{1}{2} \sum_{i=1}^{2} \sum_{n=0}^{\infty} \epsilon^n \frac{\partial_{x_i}^n \mathcal{L}(\theta)}{n!} = \log \left( 1 + e^{-\sin\left(\frac{\pi\theta}{2}\right)} \right) \tag{13}$$

$$+ \epsilon \left| \frac{\theta \cos\left(\frac{\pi\theta}{2}\right)}{1 + e^{\sin\left(\frac{\pi\theta}{2}\right)}} \right| + \epsilon^2 \operatorname{sign} \left( \theta \cos\left(\frac{\pi\theta}{2}\right) \right)^2 H(\theta) + \mathcal{O}(\epsilon^3),$$

which amounts to the result given in Eq. (6) presented in the the main text when plugging in $x_{1,2} = \pm\frac{\pi}{2}$, where we defined the sample curvature as

$$H(\theta) = \partial_{x_i}^2 \mathcal{L}(f_\theta(x_i), y_i) = \left\{ \begin{array}{c} \dfrac{\theta^2 \left( \sin(\theta x_1) + e^{\sin(\theta x_1)}(\sin(\theta x_1) + \cos^2(\theta x_1)) \right)}{2 \left( 1 + e^{\sin(\theta x_1)} \right)^2} \\[2em] -\dfrac{\theta^2 e^{\sin(\theta x_2)} \left( \left(1 + e^{\sin(\theta x_2)}\right) \sin(\theta x_2) - \cos^2(\theta x_2) \right)}{2 \left( 1 + e^{\sin(\theta x_2)} \right)^2} \end{array} \right\}, \tag{14}$$

## B    THE IMPLICIT BIAS OF SINGLE STEP AT

Here, we extend the previous discussion to a general setting, where the network is given by $\boldsymbol{f}_{\boldsymbol{\theta}_t}(\boldsymbol{x}_i)$, and the loss $\mathcal{L}(\boldsymbol{f}_{\boldsymbol{\theta}_t}(\boldsymbol{x}_i), y_i)$.

We first consider the SGD equations for the single step AT as in Algorithm 1 with $S = 1$:

$$\boldsymbol{\theta}_{t+1} = \boldsymbol{\theta}_t - \alpha \cdot \nabla_{\boldsymbol{\theta}_t} \mathcal{L}(\boldsymbol{f}_{\boldsymbol{\theta}_t}(\boldsymbol{x}_i + \boldsymbol{\delta}_t^i), y_i), \qquad \boldsymbol{\delta}_t^i = \epsilon \cdot \operatorname{sign}\left( \nabla_{\boldsymbol{x}_i} \mathcal{L}(\boldsymbol{f}_{\boldsymbol{\theta}_t}(\boldsymbol{x}_i), y_i) \right). \tag{15}$$

It is useful to define the argument of $\boldsymbol{\delta}_t^i$ as

$$\boldsymbol{g}_t^i \equiv \nabla_{\boldsymbol{x}_i} \mathcal{L}(\boldsymbol{f}_{\boldsymbol{\theta}_t}(\boldsymbol{x}_i), y_i) \rightarrow \boldsymbol{\delta}_t^i = \epsilon \cdot \operatorname{sign}(\boldsymbol{g}_t^i) \tag{16}$$

As noted in Section 4.1, since the weights are updated according to the loss gradient $\nabla_{\boldsymbol{\theta}} \mathcal{L}(\boldsymbol{\theta}, \epsilon)$, it is sufficient to study the effective loss being optimized due to the AT perturbation. In particular, we expand the perturbed loss to second order, neglecting $\mathcal{O}(\epsilon^3)$ terms, giving

$$\mathcal{L}(\boldsymbol{f}_{\boldsymbol{\theta}_t}(\boldsymbol{x}_i + \boldsymbol{\delta}_t^i), y_i) \simeq \mathcal{L}(\boldsymbol{f}_{\boldsymbol{\theta}_t}(\boldsymbol{x}_i), y_i) + {\boldsymbol{\delta}_t^i}^\top \nabla_{\boldsymbol{x}_i} \mathcal{L}(\boldsymbol{f}_{\boldsymbol{\theta}_t}(\boldsymbol{x}_i), y_i) + \frac{1}{2} {\boldsymbol{\delta}_t^i}^\top \nabla_{\boldsymbol{x}_i}^2 \mathcal{L}(\boldsymbol{f}_{\boldsymbol{\theta}_t}(\boldsymbol{x}_i), y_i) \boldsymbol{\delta}_t^i, \tag{17}$$

where $|\boldsymbol{\delta}_t^i| \le \epsilon$, justifying the expansion. Rewriting Eq. (17) in terms of $\epsilon$, we obtain a familiar form

$$\mathcal{L}(\boldsymbol{f}_{\boldsymbol{\theta}_t}(\boldsymbol{x}_i + \boldsymbol{\delta}_t^i), y_i) \simeq \mathcal{L}(\boldsymbol{f}_{\boldsymbol{\theta}_t}(\boldsymbol{x}_i, y_i)) + \epsilon \|\boldsymbol{g}_t^i\|_1 + \frac{\epsilon^2}{2} \operatorname{sign}(\boldsymbol{g}_t^i)^\top \nabla_{\boldsymbol{x}_i}^2 \mathcal{L}(\boldsymbol{f}_{\boldsymbol{\theta}_t}(\boldsymbol{x}_i, y_i)) \operatorname{sign}(\boldsymbol{g}_t^i). \tag{18}$$

Similar to the structure of Eq. (7), we see that the first term in Eq. (17) is the unperturbed loss, and the second term is an implicit $L_1$ regularization term for the sample gradient $\boldsymbol{g}_t^i$, pushing the single sample input gradients to zero. The third term seeks to minimize the single sample Hessian $\nabla_{\boldsymbol{x}_i}^2 \mathcal{L}(\boldsymbol{f}_{\boldsymbol{\theta}_t}(\boldsymbol{x}_i, y_i))$ under the product of $\operatorname{sign}(\boldsymbol{g}_t^i)$. As we explained in Section 4.1, and show graphically in Fig. 7, this term takes large negative values at $\epsilon_c$. Therefore, if $\epsilon > \epsilon_c$ for a given architecture and dataset, the optimal solution will overfit the perturbed data, leading to CO.

## C    THE IMPLICIT LOCAL LINEARITY REGULARIZATION OF TWO STEP AT

Next, we would like to compare the single-step FGSM result with the multi-step approach, focusing on the two step method, or $S = 2$ in Algorithm 1. Here, the perturbation is given by

$$\boldsymbol{\delta}_{2,t}^i = \epsilon_1 \cdot \operatorname{sign}\left( \boldsymbol{g}_t^i \right) + \epsilon_2 \cdot \operatorname{sign}\left( \nabla_{\boldsymbol{x}_i} \mathcal{L}(\boldsymbol{f}_{\boldsymbol{\theta}_t}(\boldsymbol{x}_i + \epsilon_1 \cdot \operatorname{sign}\left( \boldsymbol{g}_t^i \right)), y_i) \right), \tag{19}$$

where we allow two different perturbation magnitudes $\epsilon_1, \epsilon_2$, and demonstrate that controlling the ratio between them can effectively negate the phase transition and ameliorate CO.

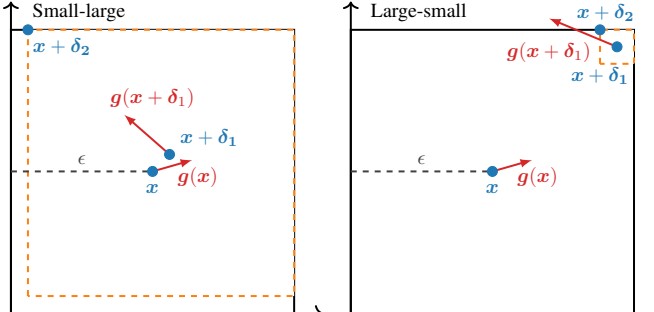

Figure 8: **Visualization of PGD-2 attacks in two dimensions:** In 🔵 we plot the trajectory of the PGD-2 attack, in 🔴 the gradients at the different points and in 🟠 the radius of the second step. Intuitively, Large-small has a very small range of motion in the second step (see 🟠) and therefore behaves like single-step AT. Alternatively, Small-large effectively regularizes the difference in gradients and curvature at $\boldsymbol{x}$ and $\boldsymbol{x} + \boldsymbol{\delta}_1$.

## C.1 First Large Step, Second Small Step

First, we consider the case of $\epsilon_1 \gg \epsilon_2$. We fix $\epsilon_1$ and expand the perturbed loss to second order in $\epsilon_2 \to 0$, leading to

$$
\begin{aligned}
\mathcal{L}(\boldsymbol{f}_{\boldsymbol{\theta}_t}(\boldsymbol{x}_i + \boldsymbol{\delta}_{2,t}^i), y_i) \approx{} & \mathcal{L}(\boldsymbol{f}_{\boldsymbol{\theta}_t}(\boldsymbol{x}_i + \boldsymbol{\delta}_{1,t}^i), y_i) \\
& + \epsilon_2 \, \mathrm{sign}\left(\nabla_{\boldsymbol{x}_i}\mathcal{L}(\boldsymbol{f}_{\boldsymbol{\theta}_t}\left(\boldsymbol{x}_i + \boldsymbol{\delta}_{1,t}^i\right), y_i)\right)^\top \nabla_{\boldsymbol{x}_i}\mathcal{L}(\boldsymbol{f}_{\boldsymbol{\theta}_t}(\boldsymbol{x}_i + \boldsymbol{\delta}_{1,t}^i), y_i) \\
& + \frac{\epsilon_2^2}{2} \, \mathrm{sign}\left(\nabla_{\boldsymbol{x}_i}\mathcal{L}(\boldsymbol{f}_{\boldsymbol{\theta}_t}\left(\boldsymbol{x}_i + \boldsymbol{\delta}_{1,t}^i\right), y_i)\right)^\top \nabla_{\boldsymbol{x}_i}^2 \mathcal{L}(\boldsymbol{f}_{\boldsymbol{\theta}_t}(\boldsymbol{x}_i + \boldsymbol{\delta}_{1,t}^i), y_i) \\
& \times \mathrm{sign}\left(\nabla_{\boldsymbol{x}_i}\mathcal{L}(\boldsymbol{f}_{\boldsymbol{\theta}_t}\left(\boldsymbol{x}_i + \boldsymbol{\delta}_{1,t}^i\right), y_i)\right),
\end{aligned}
\tag{20}
$$

where we denote $\boldsymbol{\delta}_{1,t}^i = \epsilon_1 \, \mathrm{sign}\left(\boldsymbol{g}_t^i\right)$. This effective loss corresponds to an algorithm which first takes a large step in the direction of $\boldsymbol{g}_t^i$, followed by a small step in the direction of $\nabla_{\boldsymbol{x}_i}\mathcal{L}(\boldsymbol{f}_{\boldsymbol{\theta}_t}\left(\boldsymbol{x}_i + \boldsymbol{\delta}_{1,t}^i\right), y_i)$, demonstrated graphically in the center panel of Fig. 8.

Clearly, all of the terms in Eq. (20) are evaluated at the same point, both for the sign functions and for the derivatives, which implies that this is equivalent to the single step FGSM result in Eq. (18). We therefore expect this method will undergo the same PT as a function of $\epsilon_2$, characterized by CO for small $\epsilon_2$ values. The local nature of this method implies that it is susceptible to overfitting, as the curvature evaluate for each single sample can often become large and negative. We demonstrate this clearly in the right panel of Fig. 8 for the toy model.

## C.2 First Small Step, Second Large Step

Next, we consider the alternative regime, in which $\epsilon_2 \gg \epsilon_1$. We fix $\epsilon_2$ and expand the perturbed loss to second order in $\epsilon_1 \to 0$, giving

$$
\begin{aligned}
\mathcal{L}(\boldsymbol{f}_{\boldsymbol{\theta}_t}(\boldsymbol{x}_i + \boldsymbol{\delta}_{2,t}^i), y_i) \approx{} & \mathcal{L}(\boldsymbol{f}_{\boldsymbol{\theta}_t}(\boldsymbol{x}_i), y_i) + \epsilon_1 \|\boldsymbol{g}_t^i\|_1 + \frac{\epsilon_1^2}{2}\mathrm{Tr}(\boldsymbol{H}_t^i \, \mathrm{sign}(\boldsymbol{g}_t^i)\, \mathrm{sign}(\boldsymbol{g}_t^i)^\top) \\
& + \Delta\mathcal{L}(\boldsymbol{f}_{\boldsymbol{\theta}_t}(\boldsymbol{x}_i), y_i) + \epsilon_1 \, \mathrm{sign}(\boldsymbol{g}_t^i)^\top \Delta\boldsymbol{g}_t^i + \frac{\epsilon_1^2}{2}\mathrm{Tr}(\Delta\boldsymbol{H}_t^i \, \mathrm{sign}(\boldsymbol{g}_t^i)\, \mathrm{sign}(\boldsymbol{g}_t^i)^\top),
\end{aligned}
\tag{21}
$$

where we define

$$
\Delta\mathcal{L}(\boldsymbol{f}_{\boldsymbol{\theta}_t}(\boldsymbol{x}_i), y_i) \equiv \mathcal{L}(\boldsymbol{f}_{\boldsymbol{\theta}_t}(\boldsymbol{x}_i + \epsilon_2 \, \mathrm{sign}(\boldsymbol{g}_t^i)), y_i) - \mathcal{L}(\boldsymbol{f}_{\boldsymbol{\theta}_t}(\boldsymbol{x}_i), y_i),
\tag{22}
$$

$$
\Delta\boldsymbol{g}_t^i = \Delta\nabla_{\boldsymbol{x}_i}\mathcal{L}(\boldsymbol{f}_{\boldsymbol{\theta}_t}(\boldsymbol{x}_i), y_i) \equiv \nabla_{\boldsymbol{x}_i}\mathcal{L}(\boldsymbol{f}_{\boldsymbol{\theta}_t}(\boldsymbol{x}_i + \epsilon_2 \, \mathrm{sign}(\boldsymbol{g}_t^i)), y_i) - \nabla_{\boldsymbol{x}_i}\mathcal{L}(\boldsymbol{f}_{\boldsymbol{\theta}_t}(\boldsymbol{x}_i), y_i),
$$

$$
\Delta\boldsymbol{H}_t^i = \Delta\nabla_{\boldsymbol{x}_i}^2\mathcal{L}(\boldsymbol{f}_{\boldsymbol{\theta}_t}(\boldsymbol{x}_i), y_i) \equiv \nabla_{\boldsymbol{x}_i}^2\mathcal{L}(\boldsymbol{f}_{\boldsymbol{\theta}_t}(\boldsymbol{x}_i + \epsilon_2 \, \mathrm{sign}(\boldsymbol{g}_t^i)), y_i) - \nabla_{\boldsymbol{x}_i}^2\mathcal{L}(\boldsymbol{f}_{\boldsymbol{\theta}_t}(\boldsymbol{x}_i), y_i).
$$

A complete derivation of the effective loss is provided in Appendix D, but it can be intuitively explained by adding and subtracting the unperturbed gradient and curvature terms, resulting in effective regularization terms for the gradient and Hessian differences, as well as the point-wise terms.

This method clearly differs from the one presented in Appendix C.1, as it can truly be understood as a non-local method, involving the evaluation of the Hessian and gradients at different points, and thus flattening the loss landscape, avoiding the PT and reducing the likelihood of CO. This is illustrated in the left panel of Fig. 8, where taking a small initial step and a subsequent large step does not lead to CO.

### C.3 IMPLICATIONS IN NOISE-BASED SINGLE-STEP APPROACHES

Another existing approach to avoid CO is adding random noise prior to the FGSM attack in single-step AT, e.g. Fast AT (Wong et al., 2020) and N-FGSM (de Jorge et al., 2022). These approaches can be interpreted as two-step approaches with a random first step. Given $\boldsymbol{\delta}_{1,t}^i$ sampled uniformly from $\{\boldsymbol{\delta} : ||\boldsymbol{\delta}||_\infty \leq \epsilon_1\}$, Fast AT performs an FGSM step and a projection:

$$\boldsymbol{\delta}_{2,t}^i = \operatorname*{proj}_{||\boldsymbol{\delta}||_\infty \leq \epsilon} \left( \boldsymbol{\delta}_{1,t}^i + \epsilon_2 \cdot \nabla_{\boldsymbol{x}} \mathcal{L}(\boldsymbol{f}_{\boldsymbol{\theta}_t}(\boldsymbol{x}_i + \boldsymbol{\delta}_{1,t}^i), y_i) \right) . \tag{23}$$

N-FGSM performs the same operations without the projection operator. The employed hyperparameters are ($\epsilon_1 = \epsilon$, $\epsilon_2 = 1.25 \cdot \epsilon$) for Fast AT and ($\epsilon_1 = 2 \cdot \epsilon$, $\epsilon_2 = \epsilon$) for N-FGSM. While these methods are not explicitly captured by our analysis, we believe these approaches implicitly bias the loss towards local linearity by regularizing the gradient norm at the random perturbation through Eq. (18).

## D DERIVATION OF THE TWO STEP PGD EFFECTIVE LOSS

Here, we provide additional details regarding the derivation of the results presented in Appendix C.2. Recall that the perturbation to the single sample loss function is given by

$$\boldsymbol{\delta}_{2,t}^i = \epsilon_1 \cdot \operatorname{sign}\left(\boldsymbol{g}_t^i\right) + \epsilon_2 \cdot \operatorname{sign}\left(\nabla_{\boldsymbol{x}_i} \mathcal{L}(\boldsymbol{f}_{\boldsymbol{\theta}_t}(\boldsymbol{x}_i + \epsilon_1 \cdot \operatorname{sign}\left(\boldsymbol{g}_t^i\right)), y_i)\right), \tag{24}$$

we Taylor expand for $\epsilon_1 \ll 1$, while keeping $\epsilon_2$ fixed. The expansion to second order in $\epsilon_2$ is given by

$$\mathcal{L}(\boldsymbol{f}_{\boldsymbol{\theta}_t}(\boldsymbol{x}_i + \boldsymbol{\delta}_{2,t}^i), y_i) \approx \mathcal{L}(\boldsymbol{f}_{\boldsymbol{\theta}_t}(\boldsymbol{x}_i + \epsilon_2 \operatorname{sign}(\boldsymbol{g}_t^i)), y_i) \tag{25}$$
$$+ \epsilon_1 \operatorname{sign}(\boldsymbol{g}_t^i)^\top \nabla_{\boldsymbol{x}_i} \mathcal{L}(\boldsymbol{f}_{\boldsymbol{\theta}_t}(\boldsymbol{x}_i + \epsilon_2 \operatorname{sign}(\boldsymbol{g}_t^i)), y_i)$$
$$+ \frac{\epsilon_1^2}{2} \operatorname{sign}(\boldsymbol{g}_t^i)^\top \nabla_{\boldsymbol{x}_i}^2 \mathcal{L}(\boldsymbol{f}_{\boldsymbol{\theta}_t}(\boldsymbol{x}_i + \epsilon_2 \operatorname{sign}(\boldsymbol{g}_t^i)), y_i) \operatorname{sign}(\boldsymbol{g}_t^i).$$

Due to the mismatch between the sign function arguments and the terms multiplying them, we can add and subtract terms that correspond to the first step, i.e. the implicit regularization terms of the single-step FGSM with $\epsilon = \epsilon_1$, as

$$\mathcal{L}(\boldsymbol{f}_{\boldsymbol{\theta}_t}(\boldsymbol{x}_i + \boldsymbol{\delta}_{2,t}^i), y_i) \approx \mathcal{L}(\boldsymbol{f}_{\boldsymbol{\theta}_t}(\boldsymbol{x}_i), y_i) + \Delta \mathcal{L}(\boldsymbol{f}_{\boldsymbol{\theta}_t}(\boldsymbol{x}_i), y_i) \tag{26}$$
$$+ \epsilon_1 \operatorname{sign}(\boldsymbol{g}_t^i)^\top \nabla_{\boldsymbol{x}_i} \mathcal{L}(\boldsymbol{f}_{\boldsymbol{\theta}_t}(\boldsymbol{x}_i), y_i)$$
$$+ \epsilon_1 \operatorname{sign}(\boldsymbol{g}_t^i)^\top \Delta \nabla_{\boldsymbol{x}_i} \mathcal{L}(\boldsymbol{f}_{\boldsymbol{\theta}_t}(\boldsymbol{x}_i), y_i)$$
$$+ \frac{\epsilon_1^2}{2} \operatorname{sign}(\boldsymbol{g}_t^i)^\top \nabla_{\boldsymbol{x}_i}^2 \mathcal{L}(\boldsymbol{f}_{\boldsymbol{\theta}_t}(\boldsymbol{x}_i), y_i) \operatorname{sign}(\boldsymbol{g}_t^i)$$
$$+ \frac{\epsilon_1^2}{2} \operatorname{sign}(\boldsymbol{g}_t^i)^\top \Delta \nabla_{\boldsymbol{x}_i}^2 \mathcal{L}(\boldsymbol{f}_{\boldsymbol{\theta}_t}(\boldsymbol{x}_i), y_i) \operatorname{sign}(\boldsymbol{g}_t^i),$$

where the definitions of the various additional "difference" terms are given in Eq. (22) in the main text, and repeated here for completeness:

$$\Delta \mathcal{L}(\boldsymbol{f}_{\boldsymbol{\theta}_t}(\boldsymbol{x}_i), y_i) \equiv \mathcal{L}(\boldsymbol{f}_{\boldsymbol{\theta}_t}(\boldsymbol{x}_i + \epsilon_2 \operatorname{sign}(\boldsymbol{g}_t^i)), y_i) - \mathcal{L}(\boldsymbol{f}_{\boldsymbol{\theta}_t}(\boldsymbol{x}_i), y_i), \tag{27}$$
$$\Delta \boldsymbol{g}_t^i = \Delta \nabla_{\boldsymbol{x}_i} \mathcal{L}(\boldsymbol{f}_{\boldsymbol{\theta}_t}(\boldsymbol{x}_i), y_i) \equiv \nabla_{\boldsymbol{x}_i} \mathcal{L}(\boldsymbol{f}_{\boldsymbol{\theta}_t}(\boldsymbol{x}_i + \epsilon_2 \operatorname{sign}(\boldsymbol{g}_t^i)), y_i) - \nabla_{\boldsymbol{x}_i} \mathcal{L}(\boldsymbol{f}_{\boldsymbol{\theta}_t}(\boldsymbol{x}_i), y_i),$$
$$\Delta \boldsymbol{H}_t^i = \Delta \nabla_{\boldsymbol{x}_i}^2 \mathcal{L}(\boldsymbol{f}_{\boldsymbol{\theta}_t}(\boldsymbol{x}_i), y_i) \equiv \nabla_{\boldsymbol{x}_i}^2 \mathcal{L}(\boldsymbol{f}_{\boldsymbol{\theta}_t}(\boldsymbol{x}_i + \epsilon_2 \operatorname{sign}(\boldsymbol{g}_t^i)), y_i) - \nabla_{\boldsymbol{x}_i}^2 \mathcal{L}(\boldsymbol{f}_{\boldsymbol{\theta}_t}(\boldsymbol{x}_i), y_i).$$

## E PROOFS

*Proof of Proposition 3.1.* Let $\boldsymbol{\delta}_s^i = \boldsymbol{\delta}_{s-1}^i + \frac{\epsilon}{S} \cdot \text{sign}\left(\boldsymbol{g_\theta}(\boldsymbol{x}_i + \boldsymbol{\delta}_{s-1}^i)\right)$, where the projection operator was omitted as with $\sigma = 0$ and $\alpha_s = 1/S \ \forall s = 1, \cdots, S$ the perturbations never achieve a norm greater than $\epsilon$. We can iteratively Taylor-expand:

$$f(\boldsymbol{x}_i + \boldsymbol{\delta}_s^i) \approx f(\boldsymbol{x}_i + \boldsymbol{\delta}_{s-1}^i) + \frac{\epsilon}{S} \cdot \text{sign}\left(\boldsymbol{g_\theta}(\boldsymbol{x}_i + \boldsymbol{\delta}_{s-1}^i)\right)^\top \boldsymbol{g_\theta}(\boldsymbol{x}_i + \boldsymbol{\delta}_{s-1}^i)$$

$$+ \frac{\epsilon^2}{2 \cdot S^2} \cdot \text{sign}\left(\boldsymbol{g_\theta}(\boldsymbol{x}_i + \boldsymbol{\delta}_{s-1}^i)\right)^\top \boldsymbol{H_\theta}(\boldsymbol{x}_i + \boldsymbol{\delta}_{s-1}^i) \, \text{sign}\left(\boldsymbol{g_\theta}(\boldsymbol{x}_i + \boldsymbol{\delta}_{s-1}^i)\right)$$

$$= f(\boldsymbol{x}_i + \boldsymbol{\delta}_{s-1}^i) + \frac{\epsilon}{S} \cdot \left|\left|\boldsymbol{g_\theta}(\boldsymbol{x}_i + \boldsymbol{\delta}_{s-1}^i)\right|\right|_1$$

$$+ \frac{\epsilon^2}{2 \cdot S^2} \cdot \text{sign}\left(\boldsymbol{g_\theta}(\boldsymbol{x}_i + \boldsymbol{\delta}_{s-1}^i)\right)^\top \boldsymbol{H_\theta}(\boldsymbol{x}_i + \boldsymbol{\delta}_{s-1}^i) \, \text{sign}\left(\boldsymbol{g_\theta}(\boldsymbol{x}_i + \boldsymbol{\delta}_{s-1}^i)\right) \, ,$$

which can be applied recursively from $s = S$ to $s = 1$ and summed from $i = 1$ to $n$ to obtain the desired result:

$$\sum_{i=1}^n \mathcal{L}(\boldsymbol{f_\theta}(\boldsymbol{x}_i + \boldsymbol{\delta}_S^i), y_i) \approx \sum_{i=1}^n \left[ \mathcal{L}(\boldsymbol{f_\theta}(\boldsymbol{x}_i), y_i) + \sum_{j=1}^S \left( \frac{\epsilon}{S} \left|\left|\boldsymbol{g_\theta}(\boldsymbol{x}_i + \boldsymbol{\delta}_{j-1}^i)\right|\right|_1 \right. \right.$$

$$\left. \left. + \frac{\epsilon^2}{2 \cdot S^2} \text{sign}(\boldsymbol{g_\theta}(\boldsymbol{x}_i + \boldsymbol{\delta}_{j-1}^i))^\top \boldsymbol{H_\theta}(\boldsymbol{x}_i + \boldsymbol{\delta}_{j-1}^i) \, \text{sign}(\boldsymbol{g_\theta}(\boldsymbol{x}_i + \boldsymbol{\delta}_{j-1}^i)) \right) \right] \, .$$

$\square$

*Proof of Theorem 4.1.* Our proof flows as:

i) We show that the optimal loss value is attained when $\sin(\theta \cdot (x_i + \delta_S^i)) = y_i$.

ii) We look for $\theta_k$ so that:

$$\sin(\theta_k \cdot (x_i - \rho)) = \sin\left(\theta_k \cdot \left(x_i - \rho + \frac{\epsilon_k}{S}\right)\right) = y_i \, ,$$

and form $\rho, \epsilon_k$ and $\theta_k$ depending on $S, a, b_k, k$.

iii) We show that $|\mathcal{L}(\sin(\theta_k \cdot (x_i - \rho)), y_i) - \mathcal{L}(\sin(\theta_k \cdot x_i), y_i)| \le 2 \cdot \theta_k \cdot \rho$.

Because of the symmetry of the $\sin$ function and our training points $\{(-\pi/2, -1), (\pi/2, 1)\}$, we will continue the analysis just by looking at the loss at $(\pi/2, 1)$.

Starting with $i)$, let $\mathcal{L}(f_{\theta_k}(\pi/2), 1) = -\sin(\theta_k \cdot (\pi/2 + \delta_S)) + \log\left(1 + e^{\sin(\theta_k \cdot (\pi/2 + \delta_S))}\right)$. It is easy to see that the optimal loss value is $\mathcal{L}^\star = \log(1 + e) - 1 \approx 0.3133$, by minimizing $\mathcal{L}(f_{\theta_k}(\pi/2), 1)$ as a function of $\delta_S$, where $\delta_S = \frac{\pi(2n+1) - \frac{\pi}{2}}{\theta} - \frac{\pi}{2}$.

Following with $ii)$, our goal is to obtain $\theta_k$ so that $\sin(\theta_k \cdot (x - \rho + \delta_S)) = 1$. In the case $\rho = 0$ it is trivial that, since $\frac{d}{dx} \sin(\theta_k \cdot (\pi/2)) = \theta_k \cdot \cos(\theta_k \cdot \pi/2)$, by setting $\theta_k = 1$, we obtain $\delta_1 = \text{sign}(\cos(\pi/2)) = \text{sign}(0) = 0$ and therefore $\delta_S = 0$. We study $\rho > 0$ as it is a more realistic scenario where the adversary "moves".

We are then set with the problem of finding:

$$\sin(\theta_k \cdot (\pi/2 - \rho + \delta_S)) = 1 \, .$$

To ease the analysis, we will further impose that:

$$\sin(\theta_k \cdot (\pi/2 - \rho + \delta_s)) = 1 \, , \forall s = 0, \cdots, S \, .$$

Then it is enough to look for:

$$\sin(\theta_k \cdot (\pi/2 - \rho)) = \sin\left(\theta_k \cdot \left(\pi/2 - \rho + \frac{\epsilon_k}{S}\right)\right) = 1 \, , \tag{28}$$

as the PGD perturbations at each step will just be $\delta_s = \frac{\epsilon_k \cdot s}{S}$. In order to satisfy Eq. (28), the minimal $\theta_k$ must be the one where the two maxima of the $\sin$ at $\theta_k \cdot (\pi/2 - \rho)$ and $\theta_k \cdot \left(\pi/2 - \rho + \frac{\epsilon_k}{S}\right)$ are contiguous, i.e., there is no other maximum in between them. For this to happen, we just need to find:

$$\left.\begin{array}{r} \theta_k \cdot (\pi/2 - \rho) = \pi/2 + 2 \cdot \pi \cdot k \\ \theta_k \cdot \frac{\epsilon_k}{S} = 2 \cdot \pi \end{array}\right\}, \tag{29}$$

For some $k = 0, \cdots, \infty$. Then, by setting $\epsilon_k = \frac{2 \cdot \pi \cdot S}{b_k}$ and $\rho = \frac{\pi}{2 \cdot a}$, we have from the second equation $\theta_k = b_k$. We can substitute into the first equation of Eq. (29) and solve for $b_k$:

$$b_k = \frac{1 + 4 \cdot k}{1 - \frac{1}{a}}.$$

It is then easy to see that $\sin(\theta_k \cdot (\pi/2 - \rho + \epsilon_k)) = \sin\left(\frac{1+4\cdot k}{1-\frac{1}{a}} \cdot (\pi/2 - \frac{\pi}{2\cdot a} + \frac{(2\cdot\pi\cdot S)\cdot(1-\frac{1}{a})}{1+4\cdot k})\right) = \sin\left(\pi/2 \cdot \frac{1+4\cdot k}{1-\frac{1}{a}} \cdot (1 - \frac{1}{a} + \frac{(4\cdot S)\cdot(1-\frac{1}{a})}{1+4\cdot k})\right) = \sin\left(\pi/2 \cdot (1 + 4 \cdot k + (4 \cdot S))\right) = 1$. And that similarly $\sin(\theta_k \cdot (\pi/2 - \rho + \frac{\epsilon_k}{2\cdot S})) = -1$, which constitutes an adversarial example.

Finally, we show iii):

$$|\mathcal{L}\left(\sin(\theta_k \cdot (\pi/2 - \rho)), 1\right) - \mathcal{L}\left(\sin(\theta_k \cdot \pi/2), 1\right)| \leq \rho \cdot \max_{x \in \mathbb{R}} \frac{d}{dx}\mathcal{L}\left(\sin(\theta_k \cdot x), 1\right)$$

$$= \rho \cdot \max_{x \in \mathbb{R}} \theta_k \cdot \left|\cos(\theta_k \cdot x)\left(\frac{e^{\sin(\theta_k \cdot x)}}{1 + e^{\sin(\theta_k \cdot x)}} - 1\right)\right|$$

$$[|\cos(\theta_k \cdot x)| \leq 1 \text{ and } \left|\frac{e^{\sin(\theta_k \cdot x)}}{1 + e^{\sin(\theta_k \cdot x)}} - 1\right| \leq 2] \leq 2 \cdot \rho \cdot \theta_k$$

$$= \frac{\pi \cdot b_k}{a},$$

where in the first line, we used the Taylor remainder of the perturbed loss. This concludes the proof. $\qquad\square$

*Proof of Proposition 3.2.* Let the AT problem be as in Eq. (AT):

$$\min_{\boldsymbol{\theta}_\alpha} \frac{1}{n} \sum_{i=1}^n \max_{||\boldsymbol{\delta}_i||_\infty \leq \epsilon_\alpha} \mathcal{L}(\boldsymbol{f}_{\boldsymbol{\theta}}(\alpha \cdot \boldsymbol{x}_i + \boldsymbol{\delta}_i), y_i) = \min_{\hat{\boldsymbol{\theta}}_\alpha, \boldsymbol{W}_\alpha, \boldsymbol{b}_\alpha} \frac{1}{n} \sum_{i=1}^n \max_{||\boldsymbol{\delta}_i||_\infty \leq \epsilon_\alpha} \mathcal{L}(\hat{\boldsymbol{f}}_{\hat{\boldsymbol{\theta}}_\alpha}(\boldsymbol{W}_\alpha(\alpha \cdot \boldsymbol{x}_i + \boldsymbol{\delta}_i) + \boldsymbol{b}_\alpha), y_i)$$

$$= \min_{\hat{\boldsymbol{\theta}}_\alpha, \boldsymbol{W}_\alpha, \boldsymbol{b}_\alpha} \frac{1}{n} \sum_{i=1}^n \max_{||\boldsymbol{\delta}_i||_\infty \leq \epsilon_\alpha} \mathcal{L}(\hat{\boldsymbol{f}}_{\hat{\boldsymbol{\theta}}_\alpha}(\alpha \cdot \boldsymbol{W}_\alpha(\boldsymbol{x}_i + \frac{1}{\alpha}\boldsymbol{\delta}_i) + \boldsymbol{b}_\alpha), y_i)$$

$$= \min_{\hat{\boldsymbol{\theta}}_\alpha, \boldsymbol{W}_\alpha, \boldsymbol{b}_\alpha} \frac{1}{n} \sum_{i=1}^n \max_{||\boldsymbol{\delta}_i||_\infty \leq \epsilon_\alpha/\alpha} \mathcal{L}(\hat{\boldsymbol{f}}_{\hat{\boldsymbol{\theta}}_\alpha}(\alpha \cdot \boldsymbol{W}_\alpha(\boldsymbol{x}_i + \boldsymbol{\delta}_i) + \boldsymbol{b}_\alpha), y_i)$$

$$[\boldsymbol{W} = \alpha \cdot \boldsymbol{W}_\alpha, \boldsymbol{b} = \boldsymbol{b}_\alpha, \boldsymbol{\theta} = \boldsymbol{\theta}_\alpha \text{ and } \epsilon = \epsilon/\alpha] = \min_{\hat{\boldsymbol{\theta}}, \boldsymbol{W}, \boldsymbol{b}} \frac{1}{n} \sum_{i=1}^n \max_{||\boldsymbol{\delta}_i||_\infty \leq \epsilon} \mathcal{L}(\hat{\boldsymbol{f}}_{\hat{\boldsymbol{\theta}}}(\boldsymbol{W}(\boldsymbol{x}_i + \boldsymbol{\delta}_i) + \boldsymbol{b}), y_i)$$

$$= \min_{\boldsymbol{\theta}} \frac{1}{n} \sum_{i=1}^n \max_{||\boldsymbol{\delta}_i||_\infty \leq \epsilon} \mathcal{L}(\boldsymbol{f}_{\boldsymbol{\theta}}(\boldsymbol{x}_i + \boldsymbol{\delta}_i), y_i). \tag{30}$$

This shows that performing AT in the dataset $\{(\boldsymbol{x}_i \cdot \alpha, y_i)\}_{i=1}^n$ with $\epsilon_\alpha$ is effectively the same as performing AT on the standard dataset $\{(\boldsymbol{x}_i, y_i)\}_{i=1}^n$ with $\epsilon = \epsilon_\alpha/\alpha$. $\qquad\square$

*Proof of Corollary 3.3.* As proved in Proposition 3.2, performing AT in the dataset $\{(\boldsymbol{x}_i \cdot \alpha, y_i)\}_{i=1}^n$ with $\epsilon_\alpha$ is effectively the same as performing AT on the standard dataset $\{(\boldsymbol{x}_i, y_i)\}_{i=1}^n$ with $\epsilon = \epsilon_\alpha/\alpha$.

This means that if in the standard dataset $\{(\boldsymbol{x}_i, y_i)\}_{i=1}^n$ we have $(\beta, \eta)$-CO for $\epsilon > \epsilon_c$, since rescaling the dataset results in an effective adversarial budget of $\epsilon_\alpha = \epsilon \cdot \alpha$, in the re-scaled dataset $\{(\boldsymbol{x}_i \cdot \alpha, y_i)\}_{i=1}^n$ we will have $(\beta, \eta)$-CO for $\epsilon > \epsilon_c \cdot \alpha$. $\qquad\square$

*Proof of Corollary 4.2.* Based on Definition 2.1, we need to show:

    i) Nearly perfect accuracy in the PGD perturbations $\delta_S^i$.

    ii) Close-to-zero accuracy in any other points with $|\delta_\star^i| \leq \epsilon_k$.

Since we are going to make arguments when $a \to \infty$, we will make the dependence of every variable with respect to $a$ explicit, e.g., $\theta_k(a)$. Our result will hold with $\beta = \eta = 0$ in Definition 2.1. The condition i) is obtained by construction of Theorem 4.1, where we showed that:

$$\sin\left(\theta_k(a) \cdot \left(x_i - \frac{\pi}{2 \cdot a} + \epsilon_k(a)\right)\right) = y_i \quad \forall i \in \{1, 2\}, \tag{31}$$

which means that because $\frac{d \sin(\theta \cdot x)}{dx} = \theta \cdot \cos(\theta \cdot x) \leq \theta$, by Lipchitzness:

$$|\sin\left(\theta_k(a) \cdot \left(x_i - \frac{\pi}{2 \cdot a} + \epsilon_k(a)\right)\right) - \sin\left(\theta_k(a) \cdot (x_i + \epsilon_k(a))\right)| \leq \theta_k(a) \cdot \frac{\pi}{2 \cdot a}$$

$$= \frac{1 + 4 \cdot k}{1 - \frac{1}{a}} \cdot \frac{\pi}{2 \cdot a} \quad \forall i \in \{1, 2\},$$

which implies $\lim_{a \to \infty} \sin\left(\theta_k(a) \cdot (x_i + \epsilon_k(a))\right) = y_i \quad \forall i \in \{1, 2\}$. Similarly, we can show ii). Firstly, let us see that:

$$\sin\left(\theta_k(a) \cdot \left(x_i - \frac{\pi}{2 \cdot a} \pm \frac{\epsilon_k(a)}{2 \cdot S}\right)\right) = -y_i \quad \forall i \in \{1, 2\}.$$

Starting from the definition and using the fact that $\epsilon_k(a) = \frac{2\pi S}{\theta_k(a)}$ and Eq. (31):

$$\sin\left(\theta_k(a) \cdot \left(x_i - \frac{\pi}{2 \cdot a} + \frac{\epsilon_k(a)}{2 \cdot S}\right)\right) = \sin\left(\theta_k(a) \cdot \left(x_i - \frac{\pi}{2 \cdot a} + \epsilon_k(a) \pm \frac{(2 \cdot S - 1) \cdot \epsilon_k(a)}{2 \cdot S}\right)\right)$$

$$[\epsilon_k(a) = \frac{2\pi S}{\theta_k(a)}] = \sin\left(\theta_k(a) \cdot \left(x_i - \frac{\pi}{2 \cdot a} + \epsilon_k(a)\right) \pm (2 \cdot S - 1) \cdot \pi\right)$$

$$[\sin(x \pm p \cdot \pi) = -\sin(x) \text{ for odd } p] = -\sin\left(\theta_k(a) \cdot \left(x_i - \frac{\pi}{2 \cdot a} + \epsilon_k(a)\right)\right)$$

$$[\text{Eq. (31)}] = -y_i \quad \forall i \in \{1, 2\}.$$

Then, using the same arguments as before:

$$\lim_{a \to \infty} \sin\left(\theta_k(a) \cdot \left(x_i \pm \frac{\epsilon_k(a)}{2 \cdot S}\right)\right) = -y_i \quad \forall i \in \{1, 2\},$$

which shows that the points $x_i \pm \frac{\epsilon_k(a)}{2 \cdot S}$ are classified with the wrong label. By showing i) and ii), we have shown that $(0, 0)$-CO occurs with arbitrarily small $\epsilon_k(a)$, large $S$ and with arbitrarily accurate solutions when increasing $a$. $\qquad\square$

*Proof of Corollary 4.3.* It is easy to check that if $|b_k| \leq B$ and $b_k \geq 0$, we have that:

$$\epsilon_k := \frac{2\pi S}{b_k} \geq \frac{2\pi S}{B}.$$

Then, by increasing $S$, we can increase the lower bound on $\epsilon_k$ and therefore, there will not be solutions in the form of Theorem 4.1. $\qquad\square$

# F ADDITIONAL EXPERIMENTAL VALIDATION

## F.1 IMPLICIT REGULARIZATION IN TWO-STEP AT

In Appendix C, we demonstrate that two-step AT effectively biases the loss landscape to be locally linear around the training points. To confirm this prediction in practice, we train a two-layer fully connected network with a hidden size of $1,024$ and the Swish activation[2] (Ramachandran et al., 2017). We train the network on the MNIST dataset (LeCun et al., 1998), with the 15-epoch cyclic scheduler in (Andriushchenko and Flammarion, 2020) and a learning rate of $0.2$.

We report the PGD-20 accuracy and the first terms in Eq. (22) at the beginning of every epoch, for Algorithm 1 with $S = 2$ in the small-large ($\alpha_1 = 5/255$, $\alpha_2 = 65/255$) and large-small ($\alpha_1 = 65/255$, $\alpha_2 = 5/255$) setups. Due to the computational expenses of computing the second order terms in Eq. (22), we compute these terms as the average in a small sample of 16 randomly selected points from the training set. Results are averaged over the training with 3 random seeds.

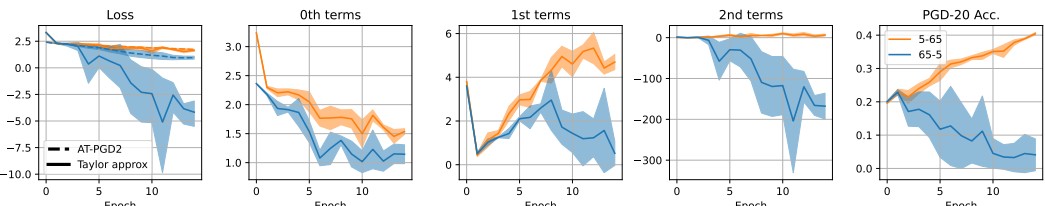

Figure 9: **Implicit regularization terms and PGD-20 adversarial accuracy in 2-step AT:** We train a two-layer fully connected neural network on MNIST at $\epsilon = 70/255$ with AT-PGD-2 in the small-large ($\alpha_1 = 5/255$, $\alpha_2 = 65/255$) and large-small ($\alpha_1 = 65/255$, $\alpha_2 = 5/255$) setups. and measure the PGD-20 adversarial accuracy and the three terms in Eq. (21). CO appears in large-small at the same time as second order terms decrease. Alternatively, small-large controls curvature and does not incur in CO.

In Fig. 9, we first see that our series expansion in Eq. (21) correctly approximates the loss curve in the small-large setup, whereas in the large-small setup, this approximation does not hold. We observe that in the large-small setup, second order terms start becoming highly negative at the point where the PGD-20 accuracy starts decreasing and CO appears. On the other hand, in the small-large setup, second order terms remain close to zero and CO does not appear. When looking at the first order terms, which are a dot product between gradients at two points (see Eq. (25)), we see that this quantity goes to zero for the large-small setup, meaning that gradients become orthogonal and local linearity is lost. This is exactly what the GradAlign scheme (Andriushchenko and Flammarion, 2020) is designed to avoid. These results indicate that:

1) The second order approximation in Appendix C holds in practice and local linearity is implicitly regularized.
2) Controlling local linearity in single-step AT, as existing methods do (Qin et al., 2019; Moosavi-Dezfooli et al., 2019; Andriushchenko and Flammarion, 2020; Abad Rocamora et al., 2024), is a good strategy to control CO.

## F.2 THE PHASE TRANSITION WITH LONGER SCHEDULES

Recent works argue longer training schedules might lead to CO (Kim et al., 2021; Abad Rocamora et al., 2024). According to our analysis this can be the case, as longer schedules might converge to the CO solutions shorter schedules did not. In Fig. 10 we find that $\epsilon_c$ is slightly larger for MNIST and SVHN and slightly smaller for CIFAR10.

## F.3 THE EFFECT OF ARCHITECTURE IN THE PHASE TRANSITION

Singla et al. (2021) argue low-curvature architectures like ResNets with the Swish activation provide better properties regarding robust overfitting. In this experiment, we evaluate the appearance of CO

---

[2]We choose the swish activation as we need our classifier to be twice differentiable for $\Delta \boldsymbol{H}_t^i$ to exist.

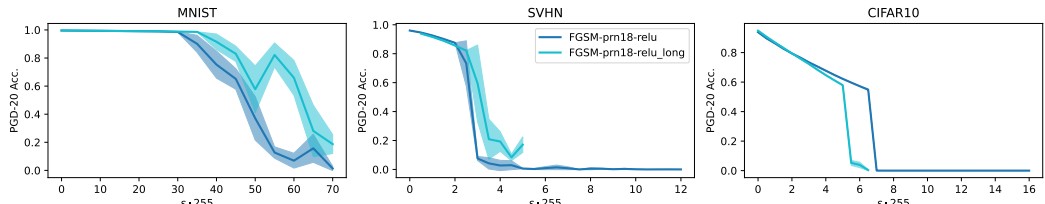

Figure 10: **Phase transition with a** 30 **or** 15 **epoch** ● **v.s. a** 200 **epoch** ● **schedule:** The phase transition occurred later for the studied $\epsilon$ in the long schedule for MNIST and SVHN, on the contrary, the phase transition occurred $1.5/255$ points earlier with the long schedule in CIFAR10.

for PreActResNet with both the ReLU and Swish activation. We additionally evaluate the performance of ViT-small (Dosovitskiy et al., 2021), which we train with the proposed training hyperparameters in Wu et al. (2022) (Sec. 5.1). We used an embedding size of $384$, a patch size of $4$, and $6$ heads.

In Fig. 11, we can observe CO occurs later for ViT-small and PRN18-Swish, confirming the result of Singla et al. (2021) for the swish activation and agreeing with our theory in Section 3.

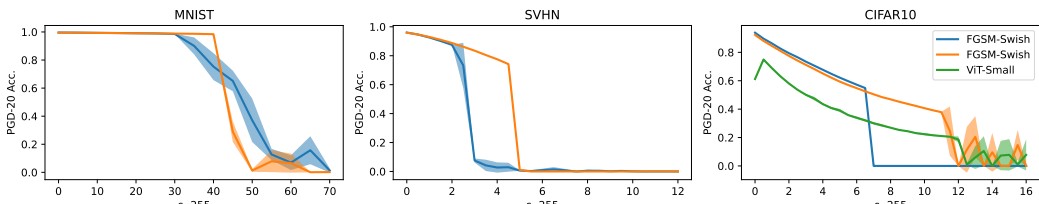

Figure 11: **Phase transition with PRN18-ReLU** ●**, PRN18-Swish** ● **and ViT-small** ●**:** The phase transition occurs later for ViT-small and PRN18-Swish, implying that architecture plays a role in the appearance of CO.

