# OpenReview forum: "A Phase Transition Induces Catastrophic Overfitting in Adversarial Training"
_ICLR.cc/2025/Conference — Submitted to ICLR 2025_

### Official Review · Reviewer_bJoQ · 2024-10-16

**Soundness:** 2
**Presentation:** 2
**Contribution:** 2
**Rating:** 6
**Confidence:** 4

**Summary:**

This paper analyzes the Catastrophic Overfitting (CO) phenomenon in Projected Gradient Descent (PGD) Adversarial Training (AT). This paper shows that a phase transition in the loss structure of as a function of the adversarial budget $\epsilon$ manifests and provides analytical and empirical evidence for the arguments by appealing to a simple model with one-dimensional inputs and a single trainable parameter. Experiments are conducted to validate the findings.

**Strengths:**

- This paper investigates the CO phenomenon and carefully analyzes the properties of CO through a toy example, which is interesting and instructive.

**Weaknesses:**

- Although interesting, the toy example used in this paper is too simplified, which is far away from what we meet in deep learning.
- Some notations are not consistent, for example, in section 2.1, the paper uses $c$ to represent the number of classes, while in section 3 $o$ is used.
- For **Proposition 3.1**, there are some issues that should be fixed.
  - In line 161, the dataset is $\\{ (x_ i, y_ i) \\}_ {i=1}^n$ but not $\\{ (x_ i, y_ i) \\}_ {i=1}^i$
  - The condition that $\alpha_ s = \frac{1}{S}$ is used in the proof, so it should be stated in the conditions in Proposition 3.1. Furthermore, such an assumption is not usually used in practice, we usually use $\alpha_ s > \frac{1}{S}$ in practice.
- **[important]** The writing is very bad, especially about the formulation of the studied problem.
  - Since the paper studies CO, the authors should define CO formally. CO is not defined although it appears many times in this paper, including the theorems. For example, in Corollary 3.3, the statements involve CO, which makes the theorem informal since CO is not formally defined. Additionally, CO is also used in Corollary 4.2 and Corollary 4.3.
  - In lines 194-195, the paper writes: "we define the perturbation threshold at which the effective loss is minimized at a high negative curvature solution as the critical $\epsilon_ c$". The expression is ambiguous, what do you mean by "at a high negative curvature solution"? How do you quantify "high"? So the definition of $\epsilon_ c$ is unclear.
  - In lines 187-189, the second question, the paper writes: "Can we have solutions where CO appears for any $S$ and $\epsilon$?" What do you mean by solutions? Does it mean PGD AT solutions in the first question? Or the FGSM solutions? This should be clarified.
  - I can not find the proof for Corollary 3.3, perhaps the reason that the authors can not provide proof for Corollary 3.3 is that the definition of CO is not clear. I think the proof of Corollary 3.3 should be included after you show a clear definition of CO. Similar problem occurs in Corollaries 4.2 and 4.3.
- I can not find the significance of Proposition 3.2. In practice, perturbating $(x, y)$ with $\delta$ where $\Vert \delta \Vert \le \epsilon_c$ is equivlent to perturbating $(\alpha\cdot x,y)$ with $\delta$ where $\Vert \delta \Vert \le \alpha \cdot \epsilon_c$. Moreover, in line 215: "With Proposition 3.2 and Corollary 3.3, we have a mechanism to re-scale the dataset and produce smaller εc that applies to modern deep architectures like ResNet and any training dataset". Yes, we can do this, but it is meaningless to simultaneously rescale $\epsilon_ c$ and $x$.

In conclusion, the paper is somewhat interesting. However, the problem is not properly formulated and some of the results lack proof. I think this paper is not prepared to be published.

**Questions:**

Please refer to the weaknesses.

---

> ### Author Response · Authors · 2024-11-23
>
> Dear Reviewer bJoQ,
>
> Thanks for your review and for helping us improve the quality of our work. We respond to your points as follows:
>
> - **P1: Although interesting, the toy example used in this paper is too simplified, which is far away from what we meet in deep learning.**
>
> That is true, the model that we employ to replicate CO is overly simplistic. But we believe that is the reason why it is a good model. The fact is that we can replicate the CO phenomena ocurring in large scales with a single parameter model. The simplicity of this model allows us to deeply analyze the problematic behaviours of AT leading to CO.
>
> - **P2: In section 2.1, the paper uses $c$ to represent the number of classes, while in section 3 $o$ is used.**
>
> Thanks for highlighting this inconsistency, we have updated the notation to use $o$ everywhere.
>
> - **P3: Typo: the dataset should be $\{x_i,y_i\}^{n} _{i=1}$ not $\{x_i,y_i\}^{i} _{i=1}$.**
>
> Thanks for pointing out this typo, it has been corrected.
>
> - **P4: $\alpha_S=1/S$ should be stated in the conditions of Proposition 3.1. In practice $\alpha_S > 1/S$ is employed.**
>
> Thanks for highlighting this issue, we have included it in the statement of Proposition 3.1
>
> - **P5: The authors should formally define CO.**
>
> Thanks for the suggestion, we have included a formal definition of CO in Definition 2.1. We believe this definition contributes to an easier understanding of our paper.
>
> - **P6: In lines 194-195, how do you quantify "high" curvature, please be more clear.**
>
> For curvature, we refer to $|\mathcal{R}_{2}|$ in Proposition 3.1, i.e., the absolute value of second order directional derivatives.
>
>
> - **P7: In lines 187-189, do you refer to FGSM solutions or AT solutions?**
>
> We refer in general when applying AT to any problem. The fact that we can show that such solutions exist in Theorem 4.1 means that for certain datasets and models, degenerate solutions with CO for any $S$ and $\epsilon$ do exist.
>
> - **P8: The proofs of the Corollaries are missing, please clearly state them.**
>
> We apologize for initially not including our proofs in the manuscript. Our corollaries are a very straightforward derivations from Proposition 3.2 and Theorem 4.1. We have included very detailed proofs in Appendix E.
>
> - **P9: What is the significance of Proposition 3.2?**
>
> We have rewritten the statement in Proposition 3.2 to be more direct and clear. The proposition simply states that training in a re-scaled dataset is equivalent to training in a standard dataset with re-scaled $\epsilon$. This has consequences as being able to produce CO for any $\epsilon$ by re-scaling the data.
>
> Please let us know if any aspect of the paper remains unclear.

---

> > ### Comment · Reviewer_bJoQ · 2024-11-25
> >
> > Thank you for the reply.
> >
> > **For P5.** the definition uses $\approx$, however, such a definition is also not clear since $\approx$ is ambiguous. Maybe a definition with an error bound is better, i.e., we define $(\alpha, \beta)$-CO if the PGD accuracy is $\ge 1-\alpha$ and the robust accuracy is $\le \beta$, where $0 \le \alpha, \beta \le 1$. Such a description is more clear than using $\approx 1$ and $\approx 0$.
> >
> > **For P6.**  Maybe you misunderstand me. My question is what value of $|\mathcal{R} _2|$ means high curvature? If $|\mathcal{R} _2| \ge \tau$ means high curvature, then what value should $\tau$ take? So the definition of high curvature is ambiguous, which means that the definition of $\epsilon _c$ is ambiguous.
> >
> > **For P9.** I read the added blue sentences in Proposition 2.2; however, they do not answer my question. The problem is that such an equivalence does not seem to be meaningful and can not provide us with more insight. In practice, perturbating $(x, y)$ with $\delta$ where $\Vert \delta \Vert \le \epsilon_c$ is equivlent to perturbating $(\alpha\cdot x,y)$ with $\delta$ where $\Vert \delta \Vert \le \alpha \cdot \epsilon_c$. For example, for a picture with an RGB range of 0 to 255, we usually transfer them into $[0, 1]$, then $x$ has range $[0,1]$. Then a perturbation with budget $\epsilon$ for $x$ and a perturbation with budget $\alpha \epsilon$ for $\alpha x$ are all equivalent to a perturbation with budget $256 \epsilon$ for the original image. Similarly, the significance of Corollary 3.3 is also not clear.

---

> > > ### Author Response · Authors · 2024-11-26
> > > **Thanks for your response**
> > >
> > > Dear reviewer bJoQ,
> > >
> > > Thanks for your response and useful suggestions. We answer to your points as follows:
> > >
> > > - **P1: The desired accuracies in CO could be defined more formally via $\alpha$ and $\beta$.**
> > >
> > > Thanks for this suggestion. We have updated the definition of CO and the Proof of Corollary 4.2 to contemplate these accuracy levels. Instead of $(\alpha,\beta)$, we employed $(\beta,\eta)$ as $\alpha$ was used for the PGD step sizes.
> > >
> > > - **P2: How high is a high curvature? How do you define $\epsilon_c$?**
> > >
> > > To improve the clarify, have added a definition of $\epsilon_c$ in Definition 2.2. Please note that $\epsilon_c$ is defined based on for which $\epsilon$ values $(\beta,\eta)$-CO appears, not on the curvature. We apologize if the definition of $\epsilon_c$ was not clear. Since the explosion in curvature is an experimental observation and our analysis does not rely on the curvature values, we believe defining how high is high curvature overcomplicates the analysis.
> > >
> > > - **P3: Regarding the significance of Proposition 3.2 and Corollary 3.3**
> > >
> > > The intuition presented by the reviewer is valid, re-scaling the perturbations and images simultaneously, does not change our perception of them. The result in Proposition 3.2 captures exactly this intuition, with the addition that solving AT on rescaled datasets and perturbations is exactly the same as solving AT in the original setting.
> > >
> > > Despite the result in Proposition 3.2 being simple and intuitive, it had not been shown before. Additionally, in the setting of this work, it is particularly interesting as it allows us to show that $\epsilon_c$ is not only an artifact arising from the AT hyperparameters, but a quantity that is scaled with the scale of the data. We have added a remark in lines 234-236.
> > >
> > > We are thankful for the discussion. Please let us know if you have any remaining concerns.
> > >
> > > Regards,
> > >
> > > Authors

---

> > > > ### Comment · Reviewer_bJoQ · 2024-11-27
> > > >
> > > > Thank you for the reply.
> > > >
> > > > Now the definition of CO and $\epsilon_c$ is clear, so the CO in the following theorems (e.g. Corollary 3.3) should be replaced by $(\beta,\eta)$-CO and it is better to describe the relationship between $\epsilon_c$ and $\epsilon_c^\alpha = \alpha  \epsilon_c$. Furthermore, $(\beta,\eta)$-CO involves robust accuracy and PGD accuracy, however, in the proofs, the authors only consider the AT (i.e. robust accuracy) but not Algorithm 1 (i.e. PGD accuracy). So the proofs should be more rigorous
> > > > .

---

> > > > > ### Author Response · Authors · 2024-11-27
> > > > > **Thanks for your acknowledgement**
> > > > >
> > > > > Dear reviewer bJoQ,
> > > > >
> > > > > Thanks for your acknowledgment. We have updated Corollary 3.3 and its proof with the new notation.
> > > > >
> > > > > Regarding the robust and PGD accuracies, please note that Theorem 4.1 is built around showing that perfect PGD loss can be obtained for arbitrarily small $\epsilon$ and arbitrarily large $S$. This covers the first need for $(0,0)$-CO: perfect PGD accuracy. For the robust accuracy, in Corollary 4.2, we show that the points $x_i \pm \frac{\epsilon_k}{2\cdot S}$ are not well clasified in the solutions given by Theorem 4.1. This covers the second need for $(0,0)$-CO: zero robust accuracy. This is shown rigorously in the proof of Corollary 4.2.
> > > > >
> > > > > Thanks again for the discussion. We appreciate the responsiveness. If you are satisfied with our clarifications, we would appreciate an increase in the rating.
> > > > >
> > > > > Regards,
> > > > >
> > > > > Authors

---

> > > > > > ### Comment · Reviewer_bJoQ · 2024-11-27
> > > > > >
> > > > > > Thank you for the reply. I have no further questions.
> > > > > >
> > > > > > After several rounds of discussions, I think the paper has improved a lot, especially with a clear description of the concepts involved and a formal description of the results. Regarding the simplicity of the toy example and the limited contribution of the rescaling results, I have no strong willingness to accept this paper, so I increase my score to "weak accept".
> > > > > >
> > > > > > Thank the authors again for the discussions.

---

> > > > > > > ### Author Response · Authors · 2024-11-27
> > > > > > > **Thanks for your increase**
> > > > > > >
> > > > > > > Dear reviewer bJoQ,
> > > > > > >
> > > > > > > Thanks a lot for helping us improve the quality of our paper and for being responsive during this rebuttal period. We remain available in case further questions appear.
> > > > > > >
> > > > > > > Regards,
> > > > > > >
> > > > > > > Authors

---

### Official Review · Reviewer_4U7J · 2024-10-21

**Soundness:** 2
**Presentation:** 3
**Contribution:** 2
**Rating:** 6
**Confidence:** 4

**Summary:**

This paper aims to uncover various mysterious properties of the Catastrophic Overfitting (CO) through investigating the implicit bias of Adversarial Training (AT). In particular, the authors design a toy example where the model has only one trainable parameter and the dataset is composed of only two data points. In this example, the authors reveal the existence of a cutoff $\epsilon_c$ such that the adversarially trained model is biased towards solution with high curvature when $\epsilon > \epsilon_c$, leading to the CO phenomenon. In addition, the authors also design several numerical experiments to support their theoretical claims.

**Strengths:**

In general, this paper is well written and organized, making understanding the main idea and logic of this paper fairly easy. The study of CO from the perspective of phase transition by showing the existence of a cutoff $\epsilon_c$ is indeed novel. Additionally, the study of the proposed toy model is comprehensive and aligns with the goal and questions raised in the introduction.

**Weaknesses:**

Despite the aforementioned strengths and advantages, I have several concerns regarding the overstatement of contribution that makes me unwilling to give a higher score and I will discuss them as follows.

- My first concern is about the insufficient discussion of the implicit bias of AT. The authors attempt to build the implicit bias of AT to link with the appearance of CO. However, Proposition 3.1 is only a Taylor expansion of the adversarial training loss: there is no explicit characterization of the “implicit bias” of AT as it is unclear what type of solution that AT will converge to. Besides, the use of Taylor expansion to the second-order actually assumes that $L(\theta)$ is at least $C^2$-smooth, which is also neglected by the authors. It is also rather odd to say that the higher order terms of a Taylor expansion could be more significant than the lower order ones (line 192-193). As this property is important for deriving the existence of a cutoff $\epsilon_c$, it is crucial to explain this point in detail, e.g., does it contradict the essence of Taylor expansion?

   On the other hand, as this paper discusses implicit bias of AT, the connection and difference between this work and related works for implicit bias of AT, e.g., Li et al, 2019; Lyu & Zhu, 2022, should be discussed. In particular, Lyu & Zhu, 2022 established the implicit bias of AT for homogeneous deep neural networks by showing that the solution converges to a KKT point of adversarial margin maximization problem. Therefore, I think the authors overstate their contribution regarding the implicit bias of AT. As Lyu & Zhu, 2022 also unified FGSM and PGD perturbations as scale invariant perturbations, I think it would be better to discuss how this property can be connected with the CO phenomenon since the implicit bias of AT is discussed more precisely there.
- My second concern is regarding the lack of connection between the proposed toy model and practical deep neural networks. Though the characterization of the proposed toy model is somewhat comprehensive, the model is far from being realistic, as it has only one trainable parameter and the dataset has only two points. Almost all the theoretical claims are made for this toy model, which, however, are not connected to any type of realistic deep neural networks. It is unclear to me what properties are special to this toy model and what conclusions can be generalized to other models and why such generalization can be made.

----
Reference

Li et al, 2019. Implicit Bias of Gradient Descent based Adversarial Training on Separable Data.

Lyu & Zhu, 2022. Implicit bias of adversarial training for deep neural networks.

**Questions:**

1. How does Proposition 3.1 characterize the properties of the converged solution? Are there any additional conditions or assumptions for making the Taylor expansion eligible?
2. When and how will the term promotional to $\epsilon^2$ become more significant than the the term promotional to $\epsilon$ as discussed in line 192-193?

   To me it is rather odd to say that the higher order terms of a Taylor expansion are more significant than the lower order ones. Please explain this point carefully.
3. Can previous results for the implicit bias of AT be connected with results in the current paper?
4. Which theoretical claims derived from the proposed toy model can be generalized to other models and why?

---

> ### Author Response · Authors · 2024-11-24
> **Rebuttal to reviewer 4U7J**
>
> Dear Reviewer 4U7J,
>
> Thanks for your careful reading of our manuscript. We address your questions and weaknesses in the following:
>
> - **P1: Authors assume that $\mathcal{L}$ is at least twice continuously differentiable. This assumption is missing.**
>
> Thank you for pointing this out. We have include this in the statement of Proposition 3.1.
>
> - **P2: [1,2] previously studied the implicit bias of AT, how does your analysis connect with theirs?**
>
> Thank you for bringing this work to our attention, they offer an interesting point of comparison with our results. In essence, these work studied the implicit bias of the solutions that a neural network trained adversarially arrive to, extending the SVM results of Soudry (2018) and others, from standard gradient descent training to the adversarial setup, finding the equivalent SVM solution in these cases. These works focus on a specific question: given a constraint (separable data, homogeneous networks), which set of equations must be satisfied by the network parameters at the infinite time limit under gradient flow dynamics, concluding that the adversarial margin must be maximized. We ask a much simpler question in a very general setting: what effective objective is being optimized by a neural network given a fixed adversarial perturbation budget. While these questions are certainly connected, our focus was geared towrd implicit regularization and a loss landscape approach, rather than the max margin formulation. We believe it is possible to rephrase our results in the SVM language. We have included this explanation as well as the references in the revised manuscript.
>
>
> - **P3: Your theoretical insights are extracted from the toy model, the connection with more practical scenarios is unclear.**
>
> While it is true that our insights are extracted from the toy model, the implicit bias/regularization analysis is valid in general, with the toy model serving as a fully tractable example in which the phenomenology can of CO as a phase transition can be seen. The empirical results given in Fig.3 and Fig.4 show that our results qualitatively extend to real world cases, while the quantitative predictions must depend on the task/architecture and data, we further discuss this point in Section 5.3.
>
>
> - **P4: How does Proposition 3.1 characterize the properties of the converged solution?**
>
> Proposition 3.1 charecterizes the effective loss being optimized, meaning that the solution must satisfy the constraints imposed by the new terms, proportional to powers of $\epsilon$. We show that both in the toy model and in real world cases, the CO solutions are the ones which satisfy these constraints by setting the Hessian to large negative values, which results from a transition from the previous minimum of the network (on the original loss) to a new, overfitting minimum on the effective loss.
>
> - **P5: When and how will the term promotional to $\epsilon^{2}$ become more significant than the the term promotional to $\epsilon$ as discussed in line 192-193?**
>
> We can approximate this point by assuming the first and second term are bounded as $R_1,R_2$ respectively, and so we can simply equate their absolute values (or traces) as $\frac{\epsilon}{S} R_1 = \frac{\epsilon^2}{2S^2} R_2$ to obtain a threshold value at $\tilde{\epsilon} = 2 R_1 S/R_2$, at this point the second term is equivalent in contribution to the first, and should be accounted for in the effective loss.

---

> > ### Comment · Reviewer_4U7J · 2024-11-26
> >
> > Thanks for the response.
> >
> > Though I still think it would be better to directly discuss how the toy model considered in this paper can be connected to the realistic models from a theoretical perspective rather than only relying on experimental results, I understand the difficulties of conducting theoretical analysis. Anyway, most of my concerns are addressed, and I will raise my score to 6.

---

> > > ### Author Response · Authors · 2024-11-26
> > > **Thanks for your response**
> > >
> > > Dear reviewer 4U7J,
> > >
> > > Thanks for your response and for increasing the score. We agree that it would be nice to be able to theoretically connect our toy model with large scale models and datasets. In our work the connection is empirical, but we hope that in future work this connection can be strengthened.
> > >
> > > Regards,
> > >
> > > Authors

---

### Official Review · Reviewer_UJp8 · 2024-10-31

**Soundness:** 2
**Presentation:** 2
**Contribution:** 2
**Rating:** 3
**Confidence:** 4

**Summary:**

This work investigates the phenomenon of catastrophic overfitting arised in the fast adversarial training (AT), where the multi-step PGD attack is replaced by a single-step PGD (also known as FGSM) to reduce AT’s training time. To explore the causes of catastrophic overfitting, the paper constructs a toy model in which the adversarial loss has a closed-form solution when adversarial examples are generated by FGSM. Through analysis of this toy model, the paper identifies a phase transition with respect to the perturbation radius $\epsilon$: when $\epsilon$ exceeds a certain threshold, the local minima of the FGSM-induced adversarial loss (referred to as the "effective loss" in this work) exhibits higher curvature and this local minima has a clearly mismatch with the minima of the adversarial loss induced by multi-step PGD. Consequently, a model that minimizes the FGSM-induced adversarial loss tends to have a high loss under multi-step PGD attacks, which the paper suggests as an explanation for catastrophic overfitting.

**Strengths:**

The paper constructs a simple one-dimensional toy model to examine catastrophic overfitting. This model enables clear visualization of the loss landscape, illustrating the effects of varying the perturbation radius on this landscape. By plotting and comparing the landscapes of two types of adversarial losses, the model provides a clear demonstration of the factors contributing to catastrophic overfitting.

**Weaknesses:**

The paper constructs a toy model to demonstrate a scenario where catastrophic overfitting provably occurs. This approach may have limited practical utility for addressing catastrophic overfitting in fast AT. A more valuable direction would be identifying the conditions under which catastrophic overfitting does not occur, allowing for improvements to fast AT by regularizing the model to meet these conditions.

Additionally, the toy model settings differ significantly from those in deep learning models, suggesting that the theoretical insights derived from the toy model may have limited applicability to real-world deep learning scenarios.



**Potential techincal issues**

The derivation of Proposition 3.1 appears to contain inaccuracies. At line 812 in the appendix, the entire derivation is based on the recurrence relation that $\delta_{s} = \delta_{s-1} + \frac{\epsilon}{S}{\rm sign}(g_{\theta}(x+\delta_{s-1}))$ where $g_{\theta}(x)=\nabla_{x}{\cal L}(f_{\theta}(x), y)$ as defined in Proposition 3.1. However, there are issues with this recursion:

1. it omits the projection operation used in AT (see Algorithm 1, step 7.)
2. when updating $\delta_{s-1}$ in PDG-AT,  the gradient should be taken w.r.t $\delta$ rather than w.r.t $x$ . Specifically, the update should use the gradient  $\nabla_{\delta}{\cal L}(f_{\theta}(x+\delta), y)$ rather than  $g_{\theta}(x+\delta_{s-1})$. The same error also appears in (Algorithm 1, step 6).



Regarding Corollary 3.3, it seems to be derived from Proposition 3.2, but the derivation is unclear.  Proposition 3.2 simply establish that  $\max\limits_{\|\delta\|\le \epsilon}{\cal L}(f_{\theta}(W(\alpha x+\delta)), y)=\max\limits_{\|\delta\|\le \hat{\epsilon}}{\cal L}(f_{\theta}(\hat{W}( x+\delta)), y)$ with $\hat{W}= \alpha W$ and $\hat{\epsilon}  = \epsilon/\alpha$ (based on the derivations at line 893 in Appendix ).  How this result leads to the claims in Corollary 3.3 is not immediately clear.



 **Writing**

The writing in the paper is not particularly reader-friendly. For instance, the insights provided by Theorem 4.1 and Corollaries 4.2 and 4.3 are not clearly explained. Additionally, the connection between the results from the toy model analysis and their implications for understanding deep learning models is not effectively conveyed.

**Questions:**

- In the loss landscape shown in the top panel of Figure 1, why is catastrophic overfitting attributed to the increased curvature of the local minima?
- In Theorem 4.1, what does $\theta^{*} _ {k}$   represent?  Why do we choose $\theta_{k}$ as $b_{k}$?
- In Corollary 4.2, why are the classification results for the points $x_{i}\pm \epsilon_{k/2S}$ considered?  How does this relate to catastrophic overfitting?
- What are the main takeaway messages of this paper? Is catastrophic overfitting attributed to that the loss landscape has local minima with high curvatures? If true, could you provide empirical evidence on deep learning models to validate this conclusion?

---

> ### Author Response · Authors · 2024-11-23
> **Rebuttal (1/2)**
>
> Dear Reviewer UJp8,
>
> Thanks for your thorough review and carefully checking the theoretical details. We respond to your concerns as follows:
>
> - **P1: "A more valuable direction would be identifying the conditions under which catastrophic overfitting does not occur"**
>
> There are many works on this direction, such as GradAlign, LLR, CURE or ELLE, that we cover in the introduction. In most works, local linearity is assumed. In this case it is impossible for CO to happen as the single-step solution is the global solution of the inner maximization problem, therefore, the network cannot overfit to only classify well the single-step adversarial examples.
>
> In this work, we address the question of understanding why CO appears. Concretely, we discover that a phase transition induces CO in single-step AT, we show that CO can ocur with arbitrarily small $\epsilon$ and provide a mechanisms to induce CO for smaller $\epsilon$ in any dataset or model by simply re-scaling the data.
>
> - **P2: "the toy model settings differ significantly from those in deep learning models, suggesting that the theoretical insights derived from the toy model may have limited applicability to real-world deep learning scenarios"**
>
> While it is true that some of our insights are extracted from the toy model, all of our insights match the experimental observations in large scale models in the literature, e.g., the sudden appearance of CO for $\epsilon$ avobe a critical value $\epsilon_c$, or the appearance of CO for multi-step AT.
>
> - **P3: Proposition 3.1 omits the projection operator in line 7 of Algorithm 1.**
>
> Note that since $\alpha = 1/S$, and $\sigma=0$, it is guaranteed that all the perturbations along the PGD attack will be inside the ball, i.e., $||\mathbf{\delta}_{s}^{i}|| _{\infty} \leq \epsilon ~~ \forall s \in [S]$, and the projection results in the identity mapping. We have clarified this in the proof of Proposition 3.1.
>
> - **P4: When updating $\mathbf{\delta}_{s}^{i}$ in PDG-AT, the gradient should be taken w.r.t. $\mathbf{\delta}$ rather than w.r.t. $\mathbf{x}$.**
>
> Please note that $\nabla_{\mathbf{x}}h(\mathbf{x} + \mathbf{\delta}) = \nabla_{\mathbf{\delta}}h(\mathbf{x} + \mathbf{\delta})$ for any differentiable function $h : \mathbb{R}^{d} \to \mathbb{R}$. We have added a remark in lines 103-105.
>
> - **P5: How is Corollary 3.3 derived from Proposition 3.2?**
>
> Intuitively, Proposition 3.2 states that performing AT in re-scaled dataset with adversarial budget $\alpha\cdot \epsilon$ is effectively the same as performing AT in the standard dataset with adversarial budget $\epsilon$. This means that every phenomenon observed with adversarial budget $\epsilon$ in the standard dataset, will be observed with adversarial budget $\alpha\cdot \epsilon$ for the re-scaled dataset. That includes CO. We have included a detailed proof in Appendix E.
>
> - **P6: The writting could be improved. Theorem 4.1 and Corollaries 4.2 and 4.3 are not clearly explained. The connection between the theory and practical implications is not clear.**
>
> We apologize for not conveying our message clearly, we have improved our writting by:
> 1. Formally defining CO in Definition 2.1.
> 2. Rewritting the statement of Proposition 3.2 and Theorem 4.1.
> 3. Including the proofs of all the Corollaries.
>
> With respect to the relationship between the insights in the small model, we believe it is clear that similarly to the toy model:
>
> 1. **Agreeing with Corollary 4.2:** There is a clear phase transition with a critical value $\epsilon_c$ for every studied dataset and various $S$ values. Leading to nearly zero PGD-20 accuracy for $\epsilon > \epsilon_c$. This is observed in practically in Figure 3.
> 3. **Agreeing with Corollary 4.3:** When the norm is constrained (Practically with weight decay), larger number of steps $S$ produce larger $\epsilon_c$. Observed for SVHN in Figure 3.
>
>
> - **P7: In the top panel of Figure 1, why is catastrophic overfitting attributed to the increased curvature of the local minima?**
>
> Please note that CO is related to a high curvature of the loss **with respect to the input**. In the top panel we simply select the solutions for both the FGSM and AT objectives. Curvature is displayed in the bottom panel, where for larger $\epsilon$, the sinusoidal classifiers obtained with FGSM present a very high frequency, i.e., curvature.
>
> - **P8: In Theorem 4.1, what does $\theta_{k}^{\star}$ represent? Why do we choose $\theta_k$ as $b_k$?**
>
> $\theta^{\star}$ is a solution that attains the minimum loss $\mathcal{L}^{⋆} = \log(1 + e) − 1 ≈ 0.3133$. As explained in the proof of Theorem 4.1, $\mathcal{L}^{⋆}$ is attained when $\sin(\theta_{k}^{\star}\cdot(x_i + \delta_{S}^{i})) = y_i$. We believe that introducing $\theta_{k}^{\star}$ in the analysis is not necessary. We have rewritten Theorem 4.1 to compare against the optimal loss value $\mathcal{L}^{⋆}$ to simplify the presentation.

---

> > ### Author Response · Authors · 2024-11-23
> > **Rebuttal (2/2)**
> >
> > - **P9: In Corollary 4.2, why are the classification results for the points $x_i \pm \frac{\epsilon_k}{2S}$ considered? How does this relate to catastrophic overfitting?**
> >
> > Note that the points $x_i + \delta_{S}^{i}$ are well classified, the points $x_i \pm \frac{\epsilon_k}{2S}$ are not. This is clearly CO as the PGD-attacked points $x_i + \delta_{S}^{i}$ are well classified, but some other points inside the $\ell_{\infty}$ ball are not.
> >
> > This phenomenon is better visualized in Figure 2 (a), where the PGD-3 attacks ($x_i + \delta_3^{i}$) are at the minimum loss value $\mathcal{L}^{\star}$, but the points in between the PGD-3 trajectory $x_i \pm \frac{\epsilon_k}{6}$ have a high loss and are not well classified.
> >
> > - **P10: What are the main takeaway messages of this paper? Is catastrophic overfitting attributed to that the loss landscape has local minima with high curvatures? If true, could you provide empirical evidence on deep learning models to validate this conclusion?**
> >
> > In order to confirm our intuition from Proposition 3.1, we have trained PreActResNet18-swish on SVHN with $S=2$ and $\epsilon \in \{4,12\}/255$. For every epoch, we measure the implicit bias terms in Proposition 3.1 ($\mathcal{R}_1$ and $\mathcal{R}_2$), the training PGD-2 accuracy and the test PGD-20 accuracy. We observe that indeed, when CO appears, curvature at the PGD trajectory notably increases.
> >
> > Thanks again for your review and for helping us improve the quality of our work. Please let us know if you have any remaining concerns.

---

> > > ### Comment · Reviewer_UJp8 · 2024-11-27
> > > **Remaining concerns and follow-up questions**
> > >
> > > Thank you for making effort to address my concerns. I have reviewed the revised manuscript and appreciate the adjustments made to improve its clarity. However, several key issues remain unresolved in the current version:
> > >
> > > **Concerns Regarding Section 3:**
> > >
> > > In line 211, the authors raised three questions:
> > >
> > > (1) *Can we have PGD AT solutions where curvature is high?*
> > >
> > > (2) *Can we have solutions where CO appears for any* $S$ *and* $\epsilon$ *?*
> > >
> > > (3) *Can we understand why such solutions do not appear in practice?*
> > >
> > >
> > >
> > > Firstly, the motivation of raising the three questions are not super clear, especially the connection between question(1), (3) with the central theme of the paper—CO in fast AT. Due to unclear motivations and relevances, the appearance of section 3 interrupts the logical flow of the paper.
> > >
> > > Secondly, the analysis for question (1) appears highly heuristic, lacking theoretical supports. The analysis for question (2) relies on a trivial construction that the training data and the perturbation radius $\epsilon$ are scaled at the same time such that the effect of reducing $\epsilon$ is cancelled by the scaling of the data $x$. For question (3), the discussion provided in Lines 238–246 is vague, leaving the main takeaway unclear.
> > >
> > > Overall, Section 3 occupies a significant portion of the paper, yet its key messages are not effectively conveyed. Furthermore, the discussion in this section lacks clear motivation and does not establish strong connections to the CO problem, which the paper aims to address. Could the authors summarize the main purpose and key takeaways of Section 3? Additionally, Section 3 feels isolated from subsequent sections. How does it connect to the rest of the paper, particularly the follow-up sections?
> > >
> > >
> > >
> > > **Concerns about dataset rescaling:**
> > >
> > > The motivation for inducing CO through dataset rescaling is also not well explained. As a result, the experimental results presented in Section 5.5 fail to provide meaningful insights.
> > >
> > >
> > >
> > > **Follow-up questions for the revised manuscript:**
> > >
> > > In Theorem 4.1, the construction of $\theta_k$ appears somewhat artificial and seems unrelated to the context of fast AT. I was expecting $\theta_k$ to represent parameters learned through fast AT, but this does not seem to be the case. Could the authors provide additional clarification on the motivation and relevance of this construction?
> > >
> > > Theorem 4.1 provides an upper bound, but it is unclear how this bound "present sufficient conditions to observe CO" as stated in Remark 4.4. Could the authors elaborate on the connection between this bound and CO?
> > >
> > > In Line 461, the authors claim that the empirical observations in Figure 3 "align with our results in the toy model despite the simplistic assumptions of Theorem 4.1 and Corollary 4.3." However, it is not clear how Theorem 4.1 and Corollary 4.3 relate to the observation that "an abrupt decay of PGD-20 accuracy to zero occurs with a small change in $\epsilon$." Could the authors clarify this relationship?
> > >
> > > At line 463, what do you mean by "Our analysis in the toy model covers the existence of a CO solution with lower loss than the robust solution for larger $\epsilon$" ?
> > >
> > > At line 465, what do you mean by " longer schedules might converge to the CO solutions shorter schedules did not."? Figure 10 in the paper indicates that on MNIST and SVHN, FGSM with shorter training epochs encountered CO at smaller $\epsilon$ values compared to FGSM with longer training epochs. This contradicts the statement at Line 465.

---

> ### Author Response · Authors · 2024-11-27
> **Thanks for your response (1/2)**
>
> Dear reviewer UJp8,
>
> Thanks for your response and the thorough feedback on the revised version of the manuscript. We answer to your remaining concerns as follows:
>
> ## Regarding the writing of Section 3
>
> We appreciate your detailed feedback about the writing of this section. We apologize if the main ideas are not easy to extract at the moment. We believe that given your feedback, it is a good idea to integrate Sections 3 and 4 together. We can first pose the three questions an then combine Sections 3 and 4, pointing at how each insight from our theory connects to the questions. Moreover, this way, the connection between the toy model and large scale results can be made more clear. However, given that there are less than 13h remaining to update the manuscript, we believe the time is not enough to provide the desired quality. We can work on making sections 3 and 4 more reader friendly for the camera ready version.
>
> Overall, leaving the readability aside, we believe the three questions we point out are covered in the paper, let us explain:
>
> - **Question (i) Can we have PGD AT solutions where curvature is high?**
>
> Proposition 3.1 and the results in Section 5.4 answer this question affirmatively both theoretically and practically. Moreover, the theoretical results in Theorem 4.1, show that this is also the case in the toy model, where curvature along the PGD trajectory is proportional to $\theta_k^{2}$. Given that $\theta_k$ is proportional to $1/\epsilon_k$, this results in higher curvature solutions the smaller $\epsilon_k$ is.
>
> - **Question (ii) Can we have solutions where CO appears for any $S$ and $\epsilon$?**
>
> As you pointed out, via re-scaling the data, Corollary 3.3 gives us a mechanism to induce CO for arbitrarily small $\epsilon$. Nevertheless, this is not our strongest result. Check that Theorem 4.1 provides analytical solutions to the AT problem in the toy model for arbitrarily small $\epsilon_k$ and arbitrarily large $S$.
>
> - **Question (iii) Can we understand why these solutions do not appear in practice?**
>
> In lines 238–246, we argue that adding noise prior the PGD attack ($\sigma > 0$) and weight decay can help avoid CO. The noise argument is an intuition arising from our theory and since it is not fully understood, it has been included in the limitations of our work. Regarding weight decay, our result in Corollary 4.3 in the toy model and the experimental results in Figure 3 confirm that: Constraining the parameter norm and increasing the number of PGD steps $S$ can avoid CO.
>
> We will make sure these aspects are covered more clearly in the revised version of the manuscript. Thanks for helping us improve the readability of the paper.
>
> ## About dataset re-scaling
>
> As mentioned in our response to Reviewer bJoQ, our target with this theoretical result and experiment, was to show that the scale of $\epsilon_c$ is not only dependent on the hyperparameters of Algorithm 1. Our result in Corollary 3.3 shows that $\epsilon_c$ can be re-scaled independently of Algorithm 1 just by re-scaling the data. This result is somewhat expected, as re-scaling the data and perturbations is perceptually invariant to the human eye. Nevertheless, we believe that the independence of Algorithm 1 and data re-scaling, is a valuable and interesting result.
>
> ## Further questions
>
> - **P1: On the construction of $\theta_k$ through Theorem 4.1**
>
> In Theorem 4.1 we characterize the possible solutions of the AT problem when solved with Algorithm 1. Nevertheless, the convergence of Algorithm 1 to one of these solutions is not guaranteed. As other local minima exist (See Figure 1 top row), convergence will depend on the initialization of the weights, the learning rate and the order of data samples seen during training. This is not covered in Theorem 4.1.
>
> - **P2: Connection between Theorem 4.1. and CO**
>
> For a connection between Theorem 4.1 and CO please check Corollary 4.2. In Theorem 4.1, the Upper bound in the optimality gap vanishes to $0$ when increasing $a$. This shows that the optimal loss is achieved. In Corollary 4.2 we show that this results in a correct classification of the PGD points and misclassification of the points $x_i \pm \frac{\epsilon_k}{2\cdot S}$. Given our characterization of CO in Definition 2.1, this is exactly $(0,0)$-CO.
>
> - **P3: What is the relationship between Theorem 4.1, Corollary 4.3 and the observations in Figure 3?**
>
> In Figure 3, the critical value $\epsilon_c$ is larger when increasing $S$, this is exactly the theoretical insight from Corollary 4.3 in the toy model and the answer to Question (iii).
>
> - **P4: "Our analysis in the toy model covers the existence of a CO solution with lower loss than the robust solution for larger $\epsilon$"**
>
> The robust solution is given by $\theta=1$ in the toy model. In Figure 1, we can see that the FGSM loss at $\theta=1$ is higher than the optimal loss $\mathcal{L}^{\star}$ attained at $\theta_{\text{FGSM}}$ for $\epsilon > \epsilon_c$.

---

> > ### Author Response · Authors · 2024-11-27
> > **Thanks for your response (2/2)**
> >
> > - **P5: Regarding short/long schedules**
> >
> > As mentioned in **P1**, the theoretical analysis in the toy model characterizes the solutions of AT solved with Algorithm 1. Nevertheless, the convergence to such solutions is not guaranteed. [1,2] observe that when training for longer, $\epsilon_c$ can sometimes be observed earlier. Our results in Appendix F.2 show that for CIFAR10, this is the case. Please check that this is not against line 465.
> >
> > ---
> >
> > Thanks again for the detailed feedback and for helping us improve the quality of our work. Please let us know if further questions appear. If you are satisfied with our responses, we would appreciate an increase in the score.
> >
> >
> > **References**
> >
> > [1] Kim et al., Understanding catastrophic overfitting in single-step adversarial training, AAAI 2021
> >
> > [2] Abad Rocamora et al., Efficient local linearity regularization to overcome catastrophic overfitting, ICLR 2024.

---

> > > ### Comment · Reviewer_UJp8 · 2024-12-01
> > > **Response to your comments (Part 1)**
> > >
> > > Thank you for your response. Regarding your comments:
> > >
> > > >- Proposition 3.1 and the results in Section 5.4 answer this question affirmatively both theoretically and practically. Moreover, the theoretical results in Theorem 4.1, show that this is also the case in the toy model, where curvature along the PGD trajectory is proportional to $\theta_k^{2}$. Given that $\theta_k$ is proportional to $1/\epsilon_k$, this results in higher curvature solutions the smaller $\epsilon_k$ is.
> > >
> > >
> > > The authors might misunderstand my original question. My concern is about the relevance and significance of the proposed question: *"Can we have PGD AT solutions where curvature is high?"* Why is this question interesting or worth investigating in the context of the paper?
> > >
> > >
> > > The authors acknowledge that, in practice, *"multi-step AT converges to have small curvature in the neighborhood of training points"* (Line 209). Analyzing the scenario where AT converges to high-curvature solutions does not align with what is observed in practice. Consequently, the theoretical analysis based on this scenario is unlikely to uncover the true reasons behind CO in AT in real-world settings.
> > >
> > > Additionally, I do not agree that the results in Proposition 3.1 can be interpreted as "answering this question affirmatively." There is a clear gap between the solutions obtained through AT and those derived by minimizing the second-order Taylor expansion of the adversarial loss. The claims made based on Proposition 3.1 are speculative and should be treated as hypotheses rather than rigorous theoretical results.
> > >
> > > >- As you pointed out, via re-scaling the data, Corollary 3.3 gives us a mechanism to induce CO for arbitrarily small $\epsilon$. Nevertheless, this is not our strongest result. Check that Theorem 4.1 provides analytical solutions to the AT problem in the toy model for arbitrarily small $\epsilon_k$ and arbitrarily large $S$.
> > >
> > > The result presented in Corollary 3.3 lacks meaningful insight, as the construction involving the rescaling of the dataset is inherently trivial. Unfortunately, this response does not address my primary concerns regarding the significance of this result.
> > >
> > > >- In lines 238–246, we argue that adding noise prior the PGD attack ($\sigma > 0$) and weight decay can help avoid CO. The noise argument is an intuition arising from our theory and since it is not fully understood, it has been included in the limitations of our work. Regarding weight decay, our result in Corollary 4.3 in the toy model and the experimental results in Figure 3 confirm that: Constraining the parameter norm and increasing the number of PGD steps $S$ can avoid CO.
> > >
> > > The techniques that "adding noise prior the PGD attack ($\sigma > 0$) and weight decay" have already been used in PGD-AT and in fast AT, but in practice fast AT equipped with these techiques still suffers from CO. Therefore the statement that " we argue that adding noise prior the PGD attack ($\sigma > 0$) and weight decay can help avoid CO." is misleading.
> > >
> > >
> > >
> > > >- Our result in Corollary 3.3 shows that $\epsilon_c$ can be re-scaled independently of Algorithm 1 just by re-scaling the data. This result is somewhat expected, as re-scaling the data and perturbations is perceptually invariant to the human eye. Nevertheless, we believe that the independence of Algorithm 1 and data re-scaling, is a valuable and interesting result.
> > >
> > > The authors themselves acknowledge that *"this result is somewhat expected,"* which underscores its triviality. As such, the result fails to provide any meaningful or novel insights.

---

> > > > ### Comment · Reviewer_UJp8 · 2024-12-01
> > > > **Response to your comments (Part 2)**
> > > >
> > > > >- For a connection between Theorem 4.1 and CO please check Corollary 4.2. In Theorem 4.1, the Upper bound in the optimality gap vanishes to $0$ when increasing $a$. This shows that the optimal loss is achieved. In Corollary 4.2 we show that this results in a correct classification of the PGD points and misclassification of the points $x_i \pm \frac{\epsilon_k}{2\cdot S}$. Given our characterization of CO in Definition 2.1, this is exactly $(0,0)$-CO.
> > > >
> > > >
> > > >
> > > > The upper bound of Theorem 4.1 is in the form of $\pi \frac{b _k}{a}$. Taking $b_k  = \frac{1+ 4k}{1- \frac{1}{a}}$,  the upper bound turns into $\pi \frac{1+ 4k}{a- 1}$ .  Corollary 4.2 states that "by increasing $a$ and $k$  we can take $\epsilon_k$ arbitrary close to zero with arbitrary accurate solution $\theta_k$":  increasing $a$ and $k$  at the same time, although drives  $b_k$ to infinity and thus drives $\epsilon_k$ to zero,  this does not make the upper bound $\pi \frac{1+ 4k}{a- 1}$ go to zero.  Therefore  $\theta_k$ is not "arbitrary accurate" as stated in Corollary 4.2. The statement of Corollary 4.2 is therefore misleading.
> > > >
> > > > Another primary concern is that the construction of $\theta_k$  appears to be artificial and far from the solutions obtained by AT. The claim that "CO exists at arbitrarily small $\epsilon$" based on the artificial construction of  $\theta_k$ lacks practical relevance and does not provide meaningful insights into the occurrence of CO in practical scenarios.
> > > >
> > > >
> > > >
> > > > >- In Figure 3, the critical value $\epsilon_c$ is larger when increasing $S$, this is exactly the theoretical insight from Corollary 4.3 in the toy model and the answer to Question (iii).
> > > >
> > > >
> > > >
> > > >  I think the observation that "an abrupt decay of PGD-20 accuracy to zero occurs with a small change in $\epsilon$." is interesting, as it highlights the existence of a critical value $\epsilon_c$ that triggers the onset of CO.
> > > >
> > > > However, as the authors have clarified, the theoretical analysis focuses solely on the relationship between $\epsilon_c$ and $S$ without explicitly addressing the existence of $\epsilon_c$ itself. This disconnect causes the theoretical analysis to diverge from the key empirical observations presented in the paper.
> > > >
> > > >
> > > >
> > > > I appreciate that the authors make efforts to answer my questions. However, multiple concerns remain unresolved, and the theoretical contributions are limited. Therefore, I believe the paper is not yet ready for publication.

---

> > > > > ### Author Response · Authors · 2024-12-02
> > > > >
> > > > > Dear reviewer UJp8,
> > > > >
> > > > > Thank you for your responsiveness during this rebuttal period. We answer to your remaining and new concerns as follows:
> > > > >
> > > > > ## Remaining concerns
> > > > >
> > > > > - **P1: Why is it interesting to analyze question (iii): "Can we have PGD AT solutions where curvature is high?" Your setup is very far away from real AT.**
> > > > >
> > > > > As the reviewer points out, multi-step AT is understood to converge to locally linear solutions and avoids CO. This is shown empirically by Andriushchenko and Flammarion in their seminal paper [1]. Nevertheless, some other works have observed CO in multi-step AT [3].
> > > > >
> > > > > Before our paper, it was not clear if CO was avoided because of an implicit local linearity regularization in multi-step AT. In this work, we show a negative result. As intuitively devised from Proposition 3.1, by increasing curvature along the PGD trajectory, we can have high curvature solutions with any $S$. This is shown theoretically in the toy model in Theorem 4.1 with a closed form solution for any $S$ and $\epsilon_k$. Additionally, we can understand that such solutions are harder to achieve the more PGD steps are involved and the more we constrain the norm of our parameters (Corollary 4.3). We apologize if our previous responses were not clear enough.
> > > > >
> > > > > - **P2: Your result on Corollary 3.3 regarding data re-scaling are trivial**
> > > > >
> > > > > As we said in our previous response, the result is intuitive and "somewhat expected". Nevertheless, let us emphasize again that $\epsilon_c$ is re-scaled linearly without being affected by any other hyper-parameter in AT.
> > > > >
> > > > > Moreover, even though the result is easy to show, we are the first to do so. While it is not our main result, we believe it suits well within the context our paper and it should be conveyed in our work.
> > > > >
> > > > >
> > > > > - **P3: Adding noise ($\sigma \geq 0$) does not help avoid CO**
> > > > >
> > > > > This is false. As the reviewer says, it is true that CO can still happen when adding noise, as observed in [1] for RS-FGSM and in [2] for N-FGSM. Nevertheless, the empirical $\epsilon_c$ for these methods is significantly larger, i.e., $\epsilon_c = 16/255$ for N-FGSM v.s. $\epsilon_c = 8/255$ for FGSM on CIFAR10 [2]. This shows that noise does help avoid CO.
> > > > >
> > > > > ## New concerns
> > > > >
> > > > > - **P4: The upper bound in Theorem 4.1 does not decay to $0$ when increasing $k$ and $a$**
> > > > >
> > > > > This statement is only true if scaling both of these quantities together, keeping the ratio fixed. However, $k$ and $a$ are independent quantities. Given a value for $k$, we can choose $a$ to obtain the desired accuracy.
> > > > >
> > > > > As the reviewer points out, the upper bound can be simplified to $\pi\frac{1+4k}{a-1}$. A simple case where this upper bound can be made zero by taking $k$ and $a$ to infinity is taking $a=k^{2}$.
> > > > >
> > > > > - **P5: The $\epsilon_c$ values at larger scales are only empirical**
> > > > >
> > > > > This is true. Unfortunately we were not able to provide a theoretical demonstration of the existence of $\epsilon_c$ for larger scales. However, in the toy model we could and we believe the empirical observations at larger scales are strong enough.
> > > > >
> > > > > Please let us know if your remaining concerns have been addressed or new ones appear, we will be happy to answer.
> > > > >
> > > > >
> > > > > **References**
> > > > >
> > > > > [1] Andriushchenko and Flammarion. Understanding and Improving Fast Adversarial Training. NeurIPS 2020
> > > > >
> > > > > [2] Abad Rocamora et al., Efficient local linearity regurlarization to overcome catastrophic overfitting, ICLR 2024.
> > > > >
> > > > > [3] He et al., Investigating catastrophic overfitting in fast adversarial training: A self-fitting perspective. arXiv, 2023.

---

### Official Review · Reviewer_vMYY · 2024-11-02

**Soundness:** 2
**Presentation:** 2
**Contribution:** 2
**Rating:** 3
**Confidence:** 4

**Summary:**

Little is known about why multi-step AT converges to locally linear solutions or which is the underlying phenomenon resulting in CO. This work fills this gap by connecting the empirical observations with a theoretical framework.

**Strengths:**

1. The authors show that a phase transition in the loss structure of as a function of the adversarial budget $\epsilon$ manifests as Catastrophic Overfitting (CO).

2. The authors show that high curvature solutions arise in the implicit bias of PGD AT.  The authors provide analytical and empirical evidence by appealing to a simple model with one-dimensional inputs and a single trainable parameter, where the CO phenomenon can be replicated.

3. The authors compute the critical value $\epsilon_c$ in single-step AT for bounded parameter norms.

**Weaknesses:**

1.Adversarial Training (AT) (Madry et al., 2018) and its variants have proven to be the most effective strategy towards achieving adversarially robust models. Where is this inference from. It is better to replace this descirbtion with “one of the most”

2.Despite the success of these methods and the efforts in understanding CO, little is known about why multi-step AT converges to locally linear solutions or which is the underlying phenomenon resulting in CO. According to this sentence, the motivation of studying underlying phenomenon resulting in CO is insufficient. Moreover, there is little logical connection before and after.

3.It is redundant to introduce the known PGD algorithm 1, if you don’t bring in additional important ideas. Besides, the initialization of perturbation is random, not just $0$.

**Questions:**

See Weaknesses.

---

> ### Author Response · Authors · 2024-11-23
>
> Dear Reviewer vMYY,
>
> Thanks for your review. At the moment your review seems incomplete, we would like to ask if the reviewer has any other concerns. Regarding the concerns raised in your review, our answer is as follows:
>
> - **P1: "Adversarial Training (AT) (Madry et al., 2018) and its variants ..." It is better to replace this descirbtion with "one of the most".**
>
> We have incorporated your suggestion in the manuscript and referenced [RobustBench](https://robustbench.github.io/).
>
> - **P2: "According to this sentence, the motivation of studying underlying phenomenon resulting in CO is insuficient. Moreover, there is little logical connection before and after."**
>
> We do not understand the point of the reviewer, Could you please be more specific?
>
> - **P3: ".It is redundant to introduce the known PGD algorithm 1, if you don’t bring in additional important ideas. Besides, the initialization of perturbation is random, not just $0$."**
>
> We believe it is necessary for our analysis to present the PGD AT algorithm. It is true that the initialization in practice is uniform as we discuss in lines 221-222 of the original submission. We have updated line 4 of Algorithm 1 to contemplate this possibility and clarified that our analysis follows with $\sigma=0$.

---

### Meta-Review · Area_Chair_1zJV · 2024-12-20

**Metareview:**

This paper studies the phenomenon of catastrophic overfitting in the context of adversarial training with a single step of PGD (FGSM.)

Pros:
- the authors have conducted a comprehensive theoretical analysis,

Cons:
- the model investigated in the paper and the techniques are 'toy models' and far from being realistic.
- The original paper was lacking clarity.

In the end I believe that the cons slightly outweighs the pros. I suggest to the authors to investigate how to extend their theory to higher dimensional model in order to connect the theory with the 'toy model' to more practical models.

**Additional Comments On Reviewer Discussion:**

Reviewer vMYY did not engage in the discussion so I disregarded their review.

Reviewer UJp8 engaged significantly in the discussion with the authors and did not raise their score.

Even if Reviewer 4U7J and Reviewer bJoQ increased their scores as the quality of the presentation of the paper improved with the discussion, the concerns regarding the theory remained: the 1D model considered in the theory is very far from any practical machine learning model.

---

### Decision · Program_Chairs · 2025-01-22

Reject